# Molecular Structure, Binding Affinity, and Biological Activity in the Epigenome

**DOI:** 10.3390/ijms21114134

**Published:** 2020-06-10

**Authors:** Balázs Zoltán Zsidó, Csaba Hetényi

**Affiliations:** Department of Pharmacology and Pharmacotherapy, Medical School, University of Pécs, Szigeti út 12, 7624 Pécs, Hungary; zsido.balazs@pte.hu

**Keywords:** histone code, DNA, RNA, post-translational modification, in vitro, in vivo, molecular modeling, docking, high throughput screening, drug design, molecular dynamics, cancer, peptide

## Abstract

Development of valid structure–activity relationships (SARs) is a key to the elucidation of pathomechanisms of epigenetic diseases and the development of efficient, new drugs. The present review is based on selected methodologies and applications supplying molecular structure, binding affinity and biological activity data for the development of new SARs. An emphasis is placed on emerging trends and permanent challenges of new discoveries of SARs in the context of proteins as epigenetic drug targets. The review gives a brief overview and classification of the molecular background of epigenetic changes, and surveys both experimental and theoretical approaches in the field. Besides the results of sophisticated, cutting edge techniques such as cryo-electron microscopy, protein crystallography, and isothermal titration calorimetry, examples of frequently used assays and fast screening techniques are also selected. The review features how different experimental methods and theoretical approaches complement each other and result in valid SARs of the epigenome.

## 1. Molecular Background of the Epigenome

According to Waddington, epigenetics is “the branch of biology which studies the causal interactions between genes and their products, which bring the phenotype into being” [1,2]. Riggs further specified epigenetics as “the study of mitotically and/or meiotically heritable changes in gene function that cannot be explained by changes in DNA sequence” [3]. Following Waddington’s proposal, Holliday [4] also refers to a switch mechanism early in development that results in a random, yet permanent and successively heritable activation of some chromosomes and deactivation of others. This evolutionary chain of definitions of epigenetics is the hallmark of a rapidly developing and indispensable approach which “provides hope that we are more than just the sequence of our genes” [5].

Epigenetics explains distinct aspects of ontogenesis in normal physiology as well as pathophysiological effects of various diseases resulted by our lifestyles and the environment that might be inheritable [6]. The effect of lifestyle factors such as nightshift working, physical activity, stressful experiences, polyphenols and phytoestrogens in food, on epigenetic modifications has been reviewed [7]. Epigenetic regulation is important in learning, memory and neurogenesis, and it plays a role in related diseases, such as depression and schizophrenia [8]. Epigenetic changes also play a role in neurological, immunological and viral diseases [9]. Cancer is one of the most frequently studied diseases in general and in epigenetics, as well. Epigenetic alterations interfere with tumor progenitor genes, increasing the likelihood of cancer and worsening its prognosis [10,11,12]. Feinberg’s study [13] highlights a specific disease, Beckwith-Wiedemann Syndrome, which is caused by epigenetic defects that are specifically linked to cancer risk in affected patients. This opens up the possibility of accepting epigenetic alterations as cause, rather than consequence of cancer. 

To capture the epigenetic mechanisms of developmental biology it is necessary to unravel how the genetic program unfolds or is modified in the case of diseases at the level of nucleosomes. This goal can be achieved by the development of structure–activity relationships based on intermolecular interactions of bio-macromolecules directing cell cycle, transcription, translation and cellular signaling pathways [14,15,16]. In this sense, precise understanding of epigenetic regulation requires atomic level determination of interactions in nucleosomes between histone proteins and DNA [17,18], readers, and writers affecting gene expression in the brain [19]. Figure 1 sketches the afore-mentioned levels of epigenetic regulation. With this molecular background in mind, epigenetics can be also considered as “the structural adaptation of chromosomal regions so as to register, signal or perpetuate altered activity states” [20].

From a structural viewpoint, epigenetic regulatory mechanisms can be classified into the following categories according to the participant bio-macromolecules [22].

Category 1. Covalent modifications of DNA play a crucial role in processes for transfer of the genetic code, like in transcription. Such modifications have been linked [23] to specific types of cancers via enzymes such as methyltransferases, acetyltransferases, and kinases. For example, DNA methylation is associated with diabetes and cancer [24]. Methylation often occurs on cytosine and is carried out by DNA methyltransferases (DNMT), DNMT3A, DNMT3B and DNMT1. This results in gene repression by modifying the recognition sites and histone binding of DNA binding proteins. The hypermethylation of TRPA1 gene occurs in people with post-herpetic neuralgia and lower back pain, and is also associated with pain symptoms, burning sensations and a decreased heat pain threshold [25]. Acetylation of DNA is also important in the pathomechanism of certain types of cancer. DNA acetylation is controlled by two enzymatic families: (1) the histone lysine acetyltransferases (KAT) and (2) histone deacetylases (HDACs).

Category 2. Covalent modifications of histones. The core histone proteins H2B, H2A, H3 and H4 are essential constituents of the chromatin. Two copies of each histone are assembled into an octamer and a DNA super-helix of ca. 146 base pairs are organized around it forming the nucleosome (Figure 1), the elementary unit of the chromatin [26,27,28,29]. Nucleosomes are connected by linker DNA, and histone H1, which induces a compact structure upon binding [30] to finally yield a high-level structure of supercoiled helices building up the chromosomes [29]. Histones, except for histone H1, have long peptide tails passing through the DNA wrap of the nucleosomes (Figure 1 and Figure 2), between the turns of the coiled DNA. A wide range of structural elements extends from the histone fold domain motifs, that are structurally conserved regions found near the C-terminus in every core histone, responsible for organizing the histones into heterodimers. These structural elements play an important role in protein–protein interactions in epigenetics [29]. The N-terminal amino acids of histones also play a significant role in the interference between the DNA superhelix and neighboring compounds [29] and hold numerous PTMs [31] (Figure 2).

A great array of PTMs of the histones creates the ’histone code’ [27,28], completing the information of the genetic code [24,27]. The histone tails pass through the DNA supercoil and their PTMs are accessible for a direct or enzyme-mediated readout [32]. Besides the effector (reader) proteins, there are also writers, erasers [33,34] and remodelers [21] working in the heart of PTM machinery of the histone code (Figure 2). While readers recognize the PTMs, writers add, and erasers delete them, respectively.

Abundant PTMs including methylation, acetylation, phosphorylation and ubiquitination mostly appear on the N-terminal linear tails of the histones. For example, lysine residues can be methylated or acetylated, and a new study [35] shows, that their lactylation is also possible, directly stimulating gene transcription from chromatin.

Different histone PTMs play various roles in normal physiology and disease pathomechanisms. PTMs have a wide variety of functions [49], by directly controlling nucleosome stability they inflict DNA repair and transcription and even influence nucleosome structure. For example, di-methylation of the 4th lysine of the histone H3 tail (H3K4me2, the location is marked with an asterisk in Figure 2) results in transcriptional activation of protein WDR5, which plays an essential role in vertebrate development [50]. (Notably, the above-abridged form of histone PTMs will be used throughout this manuscript. The abridgment includes the type of histone “H3” in the asterisk-marked example, the type and serial number of amino acid “K3” holding the PTM, and the type and count of PTM “me2”). The lack of WDR5 function results in delay of ontogenesis, by four stages of development [51]. At the same time, histone methylation is involved in the development of cancer [24] and Huntington’s disease [24]. Various enzymes modulate this unique histone code during condensation, such as histone acetylases (HAT), HDACs, histone methylases, and other histone-modifying enzymes. Similarly to histone PTMs involved in (patho)physiology, their reader, writer and eraser enzymes also play an important role in maintaining physiological functions, and in disease pathomechanisms, creating a tempting target for drug design [34,52,53,54,55]. Histone acetylation plays an important role in regulating gene activity, through influencing the stability of the chromatin [36] and is also important in diabetes, asthma, and cancer [24,56]. De-acetylation maintains immuno-physiological pathways of host defense. Accordingly, HDAC inhibitors increase susceptibility to various pathogens in vivo [53]. Histone methylation and acetylation also partake in gene expression (silencing or promotion) of cyclin-dependent kinase 5 (Cdk5) gene. The expression of neuronal protein Cdk5 is increased upon chronic cocaine administration [57] and the Cdk5-zinc finger protein transcription factors can bi-directionally regulate Cdk5 gene expression with the enrichment of their respective histone modifications. Histone H3K9/14ac increases cocaine-induced locomotor behavior, while H3K9me2 attenuates it [58]. 

Category 3. Small non-protein coding RNAs or microRNAs (miRNAs) and long non-coding RNAs (lncRNAs). The miRNAs are responsible for the direct destruction or translational repression of their target RNAs, actually being functionally equivalent to small interfering RNAs (siRNAs) [59], whose function is to promote the degradation of mRNAs or inhibit their translation. Altered miRNA expression partakes in various cancer pathomechanisms, through silencing of tumor suppressor genes. A decrease in miRNA precursor family (miR) miR-101 expression leads to increased H3K27 trimethylation, which is a silencing mutation. A decreased miR-29 expression leads to an increase in the activity of DNMT3A and DNMT3B [24], both mechanisms result in tumor suppressor gene silencing. These methyltransferases are frequently up-regulated in lung cancer, and associated with poor prognosis [60]. The lncRNAs are involved in epigenetic regulation by mediating chromatin modification and DNA methylation. They also play a role in transcriptional regulation through modifying protein-DNA interactions by binding to transcription factors to facilitate their interaction with DNA to repress or activate mRNA, and post-transcriptional regulation by mRNA processing, as well as direct protein interactions to regulate protein (post-)translational modifications [61]. 

Category 4. Transcription factors are proteins binding DNA and regulating gene expression. They can form functional communities called transcription factor networks that regulate particular genes. For example, tripartite motif-containing protein 24 (TRIM24) is node of protein interactions, a promiscuous protein, with forty-four interacting partners, has a wide variety of functions, including as a ubiquitinase, a histone reader and a co-regulator of nuclear receptor-regulated transcription [62]. TRIM24 negatively regulates p53, a tumor suppressor, interacts with NRs, and directly associates with chromatin via its plant homeodomain (PHD)-bromodomain. As TRIM24 is a node of such an extended network, it has to be regulated precisely in order to avoid severe diseases, its knockout causes hepatocellular carcinoma, yet its overexpression leads to a poor prognostic breast cancer [62]. Nuclear receptors (NRs) are also important transcription factors that regulate gene expression upon binding to the specific ligand [63]. This receptor family includes intracellular steroid hormone receptors, among others. For example, estrogens are steroid hormones that act on nuclear receptors, namely human estrogen receptor α and β (hERα,β). These receptors act as ligand-activated transcription factors, upon estrogen binding, the receptors dimerize and bind to estrogen response elements (EREs), located at the promoter site of transcriptionally active genes [63,64,65]. Interestingly, not every gene contains an ERE sequence that is regulated by ERs, which necessitates distinct modes of endocrine action. They can modulate the function of other transcription factors, through protein–protein interactions, as non-genomic actions, moreover orphan nuclear hormone receptor SF-1, can serve as a direct binding site for hERα, but not hERβ [64]. 

Category 5. Complexes of chromatin remodeling and co-regulators. Covalent modification of DNA (Category 1), like methylation is fundamental in dynamic chromatin remodeling mechanisms [66,67]. Histone PTMs (Category 2) regulate transcription via controlling transcription factor (Category 4) accessibility [68]. The activity of transcription factors can be further modulated by hundreds of their own PTMs [69]. Histone PTMs can take their effects by influencing the overall structure of the chromatin via direct regulation of inter-nucleosomal contacts and controlling higher-order chromatin folding. They can also recruit specific chromatin modifiers [70,71] or remodeling enzymes that use the energy derived from ATP hydrolysis [70]. For example, acetylation of histone H2AX, member of the histone H2A family, is carried out by KAT5 (also known as Tip60) at the position H2AXK5, promoting H2AXK119 ubiquitination and enhancing chromatin remodeling [72]. 

Co-regulators are proteins that interact with transcription factors (nuclear receptors), to activate (co-activators) or repress (co-repressors) gene activity [63]. Co-activators participate in the regulation of a chromatin remodeling process when the condensed DNA becomes accessible for transcription. The co-regulators of the reverse process are the co-repressors. Co-regulators adopt various mechanisms of action. For example, they can play a role in the regulation of nuclear receptors, potentiating the activity of the receptor by switching between inactive and active states [63]. Leucine-rich motifs are frequent structural features of co-regulator molecules interacting with the ligand-binding domains of nuclear receptors. For example, the proline-, glutamic acid-, leucine-rich protein 1 (PELP1) [73] is a potential oncogene that interacts with ER, modulating its genomic and non-genomic functions, and its expression is misregulated in breast, endometrium and ovarian cancer progression [73]. Apart from being a co-activator for ER, PELP1 exerts its function as a co-repressor through association with HDAC2 and via deacetylation activity, suppresses histone acetylation and masks core histones from histone acetyltransferase mediated acetylation [74]. 

Histone readers and writers interact with (altered) histones, as was introduced in Category 2. For example, the switching defective/sucrose non-fermenting (SWI/SNF) and chromodomain, helicase, DNA binding (CHD) families partake in chromatin remodeling by interacting with the altered histone residues [72]. The SWI/SNF proteins have multiple bromodomains, enabling them to recognize and bind acetylated histone residues [72], and also have ATPase domains, typical of chromatin remodeling factors. The CHD proteins consist of tandem chromodomain and ATPase domains incorporated in a protein complex, called nucleosome remodeling deacetylase (NURD), which shows HDAC and chromatin remodeling properties [72]. 

Histone writer and eraser proteins can also function as co-regulators [75]. For example, histone acetyltransferases can weaken the interactions between the positively charged lysine side-chains of histones and the negatively charged DNA backbone phosphate groups by attaching an acetyl group and eliminating the positive charge, functioning as a co-activator [70]. The weakened interaction between the histone core octamer and the DNA backbone leads to destabilization of the local chromatin structure, which favors transcriptional activation [70]. On the contrary, HDACs, which remove the acetyl group, leave the lysine side-chain with a positive charge. In this way, they reinforce the local chromatin architecture, and are predominantly transcriptional co-repressors [70]. 

Histone writer and eraser proteins are often parts of large multi-protein complexes, and the composition of the complexes can determine the function of the histone writer or eraser [70]. Repressor element-1 silencing transcriptional factor (REST) has a co-repressor protein CoREST. If lysine-specific-demethylase 1 (LSD1) is complexed with CoREST, it demethylates H3K4me1/2, acting as a co-repressor, and if in complex with androgen receptor it demethylates H3K9, acting as a co-activator [70]. In contrast to histone acetyltransferases, histone demethylases show a greater substrate specificity, for example LSD1 requires a positively charged N atom, resulting a substrate specificity to H3K4me1/2 [70], interestingly the demethylation of H3K4me3 requires a jumonji domain, with a radical attack mechanism [70]. 

Histone writers can be also subjected to mutations, pathologic elevation or decrease in expression. Methyltransferase, acetyltransferase and kinase enzymes recruit additional chromatin modifiers and remodeler enzymes. Mutations of such enzymes frequently occur in diseases. For example, DNMT3A enzyme is mutated in myeloproliferative diseases and myelodysplasic syndromes [76]. Genes of KDM5A, and KDM5C code lysine-specific demethylase enzymes. KDM5A is mutated in acute myeloid leukemia [23] and plays a role in breast cancer formation [77,78]. KDM5C is mutated in renal carcinoma [23] and plays a role in acute myeloid leukemia [77]. The KAT3A enzyme is mutated in acute myeloid leukemia, acute lymphoid leukemia and transitional cell carcinoma of the urinary bladder, KAT3B is mutated in colorectal, breast and pancreatic carcinomas [23,78].

Category 6. Proteins with multiple functions. Regarding the extensive intertwined nature of epigenetic regulatory mechanisms, it is fairly common that some regulatory proteins play multiple roles in the epigenome, discussed in the previous Categories, respectively. An example of TRIM24 was mentioned in Category 4. Another example, the bromodomain PHD finger transcription factor (BPTF), is a nucleosome-remodeling factor subunit protein, which functions as a transcription factor. If amplified, it is prognostic for primary breast cancer [79]. At the same time, BPTF PHD finger is a histone reader sensitive to the state of methylation of the histone tail [80], which underlines that the activity of transcription factors is modulated by PTMs. Finally, we mention a protein coded in the alpha thalassemia/mental retardation X-linked (ARTX) gene. The protein has an N-terminal ATRX-DNMT3-DNMT3L (ADD) domain that binds histone H3 tail, and a C-terminal domain that is an ATP-dependent chromatin remodeling domain. The ADD domain has a PHD and a GATA zinc finger, the latter type of domains named after the specific binding of a DNA sequence. Interestingly, the ADD domain recognizes H3K4me3 with an atypical binding pocket at the interface between the GATA and PHD fingers [81]. Binding of ADD to histone H3 is facilitated by the recognition of methylated H3K9me3 (−12.2 kcal/mol), with an almost doubled binding enthalpy if compared to unmethylated H3K9me0 (−6.1 kcal/mol), indicating that H3K9me3 recognition is an enthalpy driven process [81] (see also Section 2.2.1). Unusually, the positive charge of the trimethyllysine is not accepted by an aromatic cage, but rather only one aromatic sidechain during the recognition by ADD [81] (see also Section 2.1.2). DNMT3L protein recognizes unmethylated histone H3 tail, and induces DNA methylation by DNMT3A2, establishing methylation patterns for heritable silencing and inactivation of the X chromosome in females [82]. 

In the previous paragraphs, the molecular background of epigenetic regulatory mechanisms was briefly sketched. The discussion was limited to only a segment of the most important molecules and a few examples. Exploration of the full proteome and interactome of the epigenetic universe seems a fairly demanding mission. However, various experimental and theoretical approaches have been adopted to answer this challenge. The next Sections survey recent approaches and selected contributions to the development of structure–activity relationships (SARs) of the epigenome.

## 2. Experimental Approaches

Exploration of molecular pathomechanisms of diseases of epigenetic origin and the discovery of new drugs require the determination of molecular structure, binding, and activity. Such experimental measurements are primary resources of new data for building SARs, and are also used for validation of computational approaches [83] of structural biology and drug design (Section 3).

### 2.1. Molecular Structure

The determination of three-dimensional structures of biomacromolecules of the epigenome is necessary for the precise description of their interactions and function at the atomic level. The technical breakthrough and first protein structures solved by X-ray crystallography date back to the previous century [84,85]. The technique requires expression, purification, and crystallization of biomacromolecules at a relatively large quantity and works typically on globular structures [86] neatly packed in the crystal lattice. Nuclear magnetic resonance (NMR) spectroscopy has started to supply structures for the Protein Databank (PDB, [87]) some decades ago. Beyond a static snapshot, NMR techniques also provide atomic resolution details on molecular dynamics of various systems including intrinsically disordered proteins [88,89]. However, the maximal measurable system size in NMR (ca. 35 kDa) is smaller than that in X-ray crystallography. Since the Nobel prize in 2017 [90], cryo-electron microscopy has been highlighted as an indispensable source of atomic resolution structures of the largest biological units.

#### 2.1.1. Trends

To satisfy the above-mentioned need for establishing new SARs, numerous biomolecular structures of epigenetic importance have recently been deposited in the PDB. A quick search of the PDB results in more than four thousand entries, and more than half of these entries were deposited in the past decade. The corresponding statistics are presented in Figure 3 with a general overview of the trends of experimental structure determination in the epigenome. The statistics are based on the counts of PDB structures relevant to Categories i–iii in Section 1. In general, X-ray crystallography is the oldest and most wide-spread technique, and it has a leading role (Figure 3A) in the determination of structures of the epigenome, as well. If considering the types of biomacromolecules, histone-containing structures form the most abundant group. The number of such entries shows a dynamic increase (Figure 3B) in the past ten years. This trend reflects the growing efforts on solving the “histone code” and exploration of the effects of PTMs (Category ii in Section 1).

The above trends of statistical figures are reflected in the progress of structure determination of important biological units such as nucleosomes. As it was discussed (Figure 1 and Figure 2), nucleosomes are the core units of the chromatin, and central scenes of the epigenome. Thus, the determination of their atomic resolution structure is of utmost importance. The first X-ray crystallographic measurements of the nucleosome date back to 1984 and confirmed the disk-like shape of the core particle at a 7 Å resolution [91]. There was constant progress towards the atomic level with a resolution of 2.8 Å in 1997 [29]. In 2002, the crystallographic structure was solved at 1.9 Å [48] with the whole histone H3 protein (PDB code 1kx5, Figure 2). 

While the nucleosome was solved by X-ray crystallography, the determination of important functional assemblies, such as nucleosome-reader complex structures remained extremely challenging, and necessitated the use of cryo-electron microscopy [92,93,94] (next paragraph). In solution NMR studies [80,95], the terminal peptide tails of histones have mostly been captured in their complexes with reader proteins. Similarly, X-ray crystallographic entries contain only part of the nucleosome-reader complexes. In many cases, structures of only the reader-bound terminal peptide tails (Figure 2) of histones have been captured [94,96]. Atomic level assignation of the DNA segments and interacting histone core sequences is often missing too. 

Following the new trends of recent years, nucleosome structures have also been determined by cryo-electron microscopy [93,94]. Although cryo-electron microscopy is still not as wide-spread as X-ray crystallography (Figure 3), it helps to overcome size and shape limitations [97,98], and has received a spotlight in the past decades. In the epigenome, cryo-electron microscopy has a remarkable role in the determination of multi-molecular units, such as the above-mentioned nucleosome-reader complexes. Determination of full structure of these complexes is of particular importance for exploration of the effect of PTMs on the nucleosome and development of SARs unraveling the histone code. Cryo-electron microscopy provides pioneering examples for the solution of full nucleosome-reader complexes. 

For example, in a recent study of Wagner et al. [94] the whole triad of the DNA–histone–reader complex was solved using cryo-electron microscopy (Figure 4). The structure contains the switch/sucrose non fermentable (SWI/SNF) chromatin structure remodeling complex with a subunit called nuclear protein STH1/NPS1, which is multi-functional as a histone H4 reader, as well. The interaction of the nucleosome core histone octamer, the DNA double helix wrapped around it, the protruding histone H4 tail and the reader protein are all visible, providing indispensable details for the development of SARs. The first electron microscopic map supplied the whole complex at a 15 Å resolution, and further refinements were possible for the nucleosome. Parts of the whole complex were also rigid body fitted from other PDB structures, which included both X-ray crystallographic and cryo-EM structures. This study [94] applies complementary experimental and theoretical methods such as map alignment, rigid body fitting, homology modeling and real space refinement with secondary structure restraints to solve the complex of more than one thousand kDa molecular weight. 

Beyond the static structures discussed in the previous paragraphs, development of up-to-date SARs necessitates the exploration of molecular dynamics of biomolecules of the epigenome. Recent studies [99,100] highlight the necessity of such information even at the level of molecular design. For probing structural dynamics of the nucleosome there are appropriate experimental methods like fluorescence resonance energy transfer (FRET) and nuclear magnetic resonance (NMR) spectroscopy [101]. FRET is well suited for the investigation of dramatic conformational and compositional changes. For example, FRET grants access to the measurement of nucleosome unwrapping equilibrium [101]. The equilibrium occurs between fully wrapped and partially unwrapped states of the nucleosome, with the binding of site-specific DNA binding proteins, the equilibrium shifts towards the unwrapped state, explaining the increasing accessibility of the DNA [102].

Apart from dramatic conformational changes, subtler changes of molecular conformations also occur as a part of nucleosome dynamics. NMR allows the measurement of protein dynamics and interactions at atomic level, even is disordered regions and transient complexes. NMR experiments probe how macromolecules shift between conformational sub-states in solution [103,104]. A special type of NMR, namely methyl-transverse relaxation optimized NMR (methyl-TROSY) is more suitable for the investigation of subtle dynamic changes, which is more typical for the histone tail-reader protein interactome [101]. The NMR linewidths of the base imino protons of DNA provide an informative insight into base pair opening dynamics [105]. A broader line shows reduced base pair stability and increased base pair opening rates, for example, oxidation of guanine led to line broadening of guanine base imino protons, and methylation of cytosine resulted in imino proton line narrowing [105], indicating that cytosine methylation stabilizes the DNA duplex. Solid state NMR can determine the binding sites on the nucleosome surface, and demonstrate the dynamic nature of the N and C terminal tails of histones within the core octamer [106]. These terminal ends are DNA bound and rich in PTMs, their structural dynamics is exploitable by histone reader, writer and eraser proteins [106]. Solid-state NMR does not have an intrinsic size limit, larger chromatin substrates can also be accessed, and is a complementing method to cryo-EM and X-ray crystallographic settings, when the plasticity of histones is thought to play a role or smaller proteins are observed [106].

Approaches combining different techniques such as small angle X-ray scattering, solution NMR spectroscopy and molecular dynamics were successfully applied to study the ubiquitin-like, containing PHD and RING finger domains, 1 (UHRF1) protein and its tandem Tudor domain–plant homeodomain (TTD-PHD) histone reader module [107]. UHRF1 is expressed in various cancers, being a promising target in antitumor therapy, a known small molecule acts by binding to its TTD. The study identified a novel antagonistic approach to UHRF1 function, through the allosteric disruption of the co-operative binding mode of its TTD-PHD module [107].

#### 2.1.2. Challenges

In addition to the large size of nucleosomal assemblies discussed in the previous Sections, structure determination faces other challenges due to complexity of the histone code system, conformational and functional diversity of the epigenome. Both experimental and theoretical (Section 3) approaches face the challenges described in detail in the forthcoming Sections.

##### The Size and Complexity of the Histone Code System

The histone code originates from PTMs on amino acids of histone proteins (Category ii, Section 1). It is even possible, that every single amino acid of a histone tail has a specific meaning and place in a peculiar vocabulary [36]. The code system has astronomical proportions if considering the large number amino acids and types of modifying groups involved (Figure 2). The number of possible codes can be illustrated using a specific case of methylation of H3 lysine residues. Histone H3 is known to be methylated at nine lysines, K4 [31,36,37,38,39,40], K9 [31,37,38,39,40], K14 [31,38], K18 [38,46], K23 [31,38,40], K27 [31,36,37,38,39,40], K36 [31,37,38,39,40], K56 [38,45] and K79 [31,37,38,39,40,46] (Figure 2). A single lysine side-chain can accept a maximum of three methyl groups, and there is the non-methylated, native amino acid resulting in four possible marks per residue. This means 49 (262,144) possible variations, only for lysine methylation of histone H3 not including, e.g., lysine ubiquitination, acetylation, arginine methylation, PTMs on serine, tyrosine, and other histones. Thus, the amount of PTMs of the histone code is almost uncountable, involved in many, if not all, DNA-templated processes [36,108]. Recent studies usually accumulate more variations than older works, highlighting that exploration of new PTMs is still an evolving field of epigenetics.

Besides its enormous size, the code system is further complicated by the yet unpredictable distribution of the different PTM types. Some amino acids like K4 even tend to accept multiple modification marks, resulting in different binding schemes. There are PTMs, missing from some histone types. Moreover, there are different histone reader, writer and eraser proteins, that can assess these altered amino acids in a wide variety of conformational possibilities. Lysine methylation, acetylation and ubiquitination appear from the 4th position up to the 123rd position on the histone chain, arginine methylation mostly occurs on the lower positions of the tail, while serine and tyrosine phosphorylation is typically closer to the N-terminal end in histone H3 and further in H2AX [40].

The complexity of the histone code is further increased by networking and cross-talk of the codes [109]. Some networking modifications enhance, while others inhibit the functions of others [40,110]. One study [70] proposes five mechanisms of PTM cross-talk, to which an extra level of complexity is ascribed over the histone code, for fine tuning of the overall control of the chromatin structure. Lysine residues might be target of various modifications, such as acetylation, methylation or ubiquitination and these agents might compete with each other. In Saccharomyces cerevisiae, methylation of H3K4 is dependent on the ubiquitination of H2BK123. The phosphorylation of H3S10 disrupts the binding of heterochromatin protein 1 (HP1) to H3K9me2/3, which would occur in the absence of the phosphorylation. In yeast, the FK506-binding protein 4 (scFpr4) proline isomerase catalyzes the interconversion of the peptide bond of H3P38, which interferes with the methylating ability of histone-lysine N-methyltransferase, H3 lysine-36 specific (scSet2) on H3K36. Finally, PHD finger protein 8 (PHF8) binds to H3K4me3 with its PHD finger, and this interaction is stronger when H3K9ac and H3K14ac occur at the same time [70]. Apart from these subtle mechanisms, PTMs also play a role in chromatin remodeling by altering the physico-chemical properties of the nucleosome. As it was discussed in Section 1, acetylation marks on histones H3 and H4 weaken DNA-histone interactions, enhances formation of accessible DNA, and transcriptional activation [41,70,111]. This phenomenon is mostly additive, with the more acetyl marks on the histone, the DNA becomes more accessible. Simultaneous acetylation of H4K79 and H3K122 has an amplified destabilizing effect on the nucleosome when compared to a single acetylation mark [41]. This means that additional PTMs, acting as a network, help each other to pose regulatory effects on the nucleosome.

The above-discussed extremely large size and high complexity of the histone code system is based on numerous corresponding complexes of the participant macromolecules (DNA, histones, effectors, etc.) at the atomic level. Experimental structure determination of such an infinite number of complexes would be an impossible undertaking, even with high throughput methods [112]. As experimental structure determination methods can clear up only tiny pathways in this jungle of the epigenome, involvement of fast, complementary theoretical approaches is necessary to speed up the exploration of new structures (Section 3).

##### Conformational Diversity and Water-Mediated Weak Interactions

Macromolecules of the epigenome, especially linear peptide tails of histones and RNAs [113] often adopt various binding conformations imposing further challenges on structure determination methods. Histone tails are linear structures, which are seldom compatible with X-ray crystallographic approaches [114] since these experimental methods are better at handling globular structures that can be crystallized [114,115]. Such linear peptides are better accessed by X-ray crystallography, when they are a part of a globular structure, like a nucleosome. In this case, the structure of the whole histone tail can be assigned [48]. Similar to histone tails, RNAs of the epigenome also pose a challenge for both X-ray crystallographic and solution NMR methods, due to their great flexibility. Size is also a limiting factor in their case [116,117]. The highly complex interactome (see previous Section) around the chromatin involves dynamic and flexible parts, exacerbating the difficulty of unraveling the machinery [112,118]. These difficulties often lead to experimental structures of complexes with N-terminal histone peptides of only 10–15 amino acids [81,95]. If the length of a histone tail exceeds 30 amino acids, and peptides of this length tend to loosely stick to the surface and give response signals that are non-specific for the original peptide [119]. In these cases, the N-terminal end of the peptides usually hang out to the bulk, and do not have any (specific) interactions with the partner protein (reader, Figure 5). This situation is experimentally challenging, as the peptide tails have dynamically changing positions, great flexibility, and conformational uncertainty as a result of their interactions with bulk water molecules in continuous thermal motion.

The histone H3–autoimmune regulator protein (AIRE) complex is a representative example of the above-mentioned situation. Most of the interactions take place between the first five amino acids of the N-terminal tail of the histone hampering the determination of the rest of the histone tail either experimentally or theoretically [99]. Histone recognition by a bromo-, homeo-, chromodomains or transcription factors involves only a shallow depression of the protein surface [91,96,97,98]. These flat binding surfaces often result in low interaction energies at most of the binding residues. As another example, the complex of the UHRF (see also Section 2.1.1) protein PHD finger and histone H3 N-terminal peptide tail is featured in Figure 5. The atomic resolution structure ([120], Figure 5A) and the corresponding [99] distribution of per-residue interaction energy values (Einter, Figure 5B) show that starting from the sixth amino acid, physical interactions tend to cease. This is a challenging situation in the investigation of histone tail binding to reader proteins both experimentally and theoretically. From an experimental perspective, the non-interacting part of the histone peptide tail moves dynamically in the bulk solvent, and it is hard to capture. On the other hand, fast computational docking approaches try to find the bound peptide position with the best possible interaction energy, creating non-existent interactions (mis-docked conformations, see also Section 3).

As a consequence of the shallow binding surfaces and few contact histone residues, the total binding affinities of (modified) histone tails are often limited to a micromolar range [121,122,123,124,125], indicating relatively weak complexes. 

Histone–partner interactions are also affected by structural water molecules located in the binding interface. However, the determination of hydration structure is challenging in many cases [126,127]. There is often a water network formed in the interface increasing the complexity and stability of the interactions. Disruption of the hydration network can lead to complex instability. For example, the interaction of the histone reader death-associated protein 6 (DAXX) to histone H3.3 N-terminal peptide tail was investigated with a special highlight on interacting water molecules [128] and their networking [127]. It was found experimentally that if one water molecule was displaced from the interfacial hydration network by introducing an active site mutation, the binding affinity was reduced by 50%. A computational investigation [127] further analyzed the networking of interfacial water molecules. The complete interfacial hydration networks were produced, using a molecular dynamics (MD)-based determination of the complete hydration structure by MobyWat [126]. In the mutant structures, important water nodes changed their positions or disappeared from the static core of the hydration network of the wild type. In agreement with the experimental results [128], the networking study [127] found that in the mutant system, the static core of the interfacial hydration network has disintegrated into a dynamic hydration network, explaining the reduced binding affinity.

##### Functional Diversity of the Histone Code

Histone reader, writer and eraser proteins are often promiscuous [62,70], their substrate specificity may depend on the complex they participate in [70]. Histone reader proteins are structurally diverse including plant homeo-, chromo-, bromodomains, Tudor, ADD, WD40 and PWWP modules [40]. On the other hand, a histone holding a PTM variation (a code) can also interact with multiple readers. The PTMs of the H3K4 residue is recognized by fourteen different reader proteins [40], including PHD finger containing proteins, recombination activating gene protein 2 (RAG2), inhibitor of growth protein 2 (ING2), BPTF (see in Category 6 of Section 1), AIRE, Tudor domain containing protein, SAGA complex associated factor 29 (Sgf29) and chromo domain containing proteins Jumonji domain containing 2A (JMJD2A) and chromodomain helicase DNA-binding (CHD) [129,130]. Another example is H3K9me3, which is regarded as a general transcriptional repressive mark [40], influencing a wide variety of cellular functions. Therefore, associating a distinct function with a PTM is challenging.

The functional diversity of the histone code is further increased by chromatin-associated protein complexes often containing multiple domains with different functions, as the same protein complex can include both a reader and a writer domain. For example, nucleosome acetyltransferase for H4 (NuA4) and Saccharomyces cerevisiae reduced potassium dependency 3 small (Rpd3S) protein, a HDAC share the same chromodomain containing subunit Esa1p-Associated Factor (Eaf3). Whereas Eaf3 is a histone reader domain [40] identically present in both NuA4 and Rpd3S complexes, NuA4 also has a histone writer domain, and Rpd3S contains a histone eraser domain, as well. 

Investigations of the effects of histone PTMs are complicated by their different accessibility [40] as a free peptide or under physiological conditions, embedded in the nucleosome. For example, in the H3K79me-reader interaction, flanking residues of H3K79 take different positions in their nucleosome-bound and free states. When wrapped in the nucleosome, there are structural constraints of the flanking residues, hindering the recognition of histone peptides by reader proteins [40]. 

Lysine acetylation PTMs are recognized by bromodomains with wide pockets, and by tandem PHD fingers with shallow binding pocket. Other residues surrounding these PTMs tend to form less characteristic contacts with the surface, resulting in a decreased substrate specificity [40]. On the other hand, recognition of lysine methylation by histone (de)methylases require higher substrate specificity [40]. Addition of a methyl mark to the lysine residue results in a positive charge, and increases hydrophobicity at the same time, which can be recognized by an aromatic cage (Figure 6C). Thus, binding surfaces of lysine methylation marks are similar to each-other [40]. At the same time, the non-methylated state of a lysine residues also acts as a coding variant as methylated lysine residues are not recognized by readers specific for non-methylated lysines, and vice versa [40]. 

For example, the PHD finger of AIRE protein binds a non-methylated lysine residue at position H3K4me0 [130] (Figure 6A). With three methyl groups attached to the K4 side-chain (H3K4me3), the binding completely diminishes according to ITC measurements [130]. In the case of AIRE PHD finger the recognition occurs on a flat binding surface of the protein, and this PHD finger does not have an aromatic cage. On the contrary, the PHD finger of the BPTF protein has an aromatic cage (Figure 6B,C), which can recognize the positively charged, trimethylated lysine residue of H3K4me3 PTM [80] (see another example in Category 6 of Section 1). This example nicely illustrates the binding diversity of histones, as the same PTM (H3K4me3) has radically different binding affinity to different readers.

### 2.2. Binding Affinity and Biological Activity

Epigenetic events manifest themselves as (patho)physiological activities at a systemic level. Structural results reviewed in the previous Sections demonstrated that formation of complexes of two or more molecules provides the background of such activities at an atomic level. Complex formation assumes that the partners have several intermolecular interactions (Figure 5) and high binding affinity to each other. Whereas structural description of intermolecular interactions is rather challenging and costly (Section 2.1.2) at an atomic level, measurement of binding affinity or in vitro activity is often less demanding, especially in the cases of routine assays and kits briefly mentioned in the next Sections. However, the large size of the epigenetic interactome, especially of histone PTMs (Section 2.1.2) indicates that there are hardly enough resources to measure all corresponding in vitro affinities. The highest level, in vivo activity measurements are again rather expensive, require special conditions of animal keeping and often limited by ethical concerns, as well. 

#### 2.2.1. Binding Affinity

Isothermal titration calorimetry (ITC) measurements are often performed [131,132,133,134] to gain an insight into the effect of histone PTMs and different states of methylation or acetylation on the side of the ligand. Besides investigation of PTMs, ITC can supply binding thermodynamics parameters for any binary system [98] via the measurement of the generated or absorbed heat during the titration of the solution of one partner with the other. ITC is a gold standard for determination of the full thermodynamics profile of complex formation including binding free energy (ΔG), enthalpy (ΔH), entropy (ΔS), and the stoichiometry (n) of the complex. Equilibrium dissociation constant (Kd) can be obviously calculated from ΔG and temperature data, as well.

ITC is a primary tool for finding new ligands and optimization of lead compounds in drug design. Appropriate (large negative) ΔG of ligand to a target molecule is a necessary, yet not sufficient requirement for pharmacological efficacy. Different targets might require distinct thermodynamic binding profiles to show biological effect upon interaction with their ligands [135]. During ligand optimization enthalpy and entropy-based approaches are applied [136,137,138,139] during the early stage of the optimization. An enthalpy excess can be introduced by additional hydrogen bonds to the interaction, while entropic optimization typically occurs during the later stages by for instance rigidifying the ligand in a bound conformation [135]. HIV reverse transcriptase inhibitors (e.g., etravirine) were subjected to entropic optimization, to avoid viral resistance upon mutational changes, by allowing a high residual mobility to be able to acquire multiple binding modes [135,137,138]. The design of high-affinity adaptive inhibitors can be achieved through engineering their vital interactions for affinity and specificity with conserved regions of the target. In addition, at moieties of the ligand that will most likely face rapidly mutating sites of the virus, flexible asymmetric mutations are introduced [138]. In these cutting-edge strategies, ITC is an indispensable technique, providing both ΔH and ΔS components of the overall binding affinity (ΔG).

In the context of epigenetics, ITC is an excellent tool to study the interactions of wild type (see also in Category 6 of Section 1) and mutated readers such as the PHD finger of AIRE and different PTM states of the N-terminal histone tail [95,130,140]. ITC investigations of such studies provide the full binding thermodynamics profile (ΔG, ΔH, ΔS, n), and allow measurement of effects of PTMs. For example, AIRE does not have an aromatic cage to accept H3K4 methylated lysine binding (Figure 6A); in agreement with this, H3K4me0 bound with the greatest affinity, and trimethylation of H3K4me3 caused a lack of binding to the AIRE reader protein [130]. A common way to perform a mutagenesis assay is to change the residues of interest to inert alanine residues (alanine scan), either on the side of the ligand or the target. For example, if the binding affinity of H3K4me0 to AIRE-PHD1 is compared to the binding affinity after the mutation of an aspartic acid residue to an alanine at the binding site of the target, the measured ΔH_o_ of binding of the same H3K4me0 is halved [130]. Beyond local changes on ligand binding, mutations of amino acids may alter function of the target protein by changing its overall integrity and global folding, as well.

In another study, [55], the binding of H3K4me3 was also investigated by ITC analysis. Binding of H3K4me3 and H3c4me3 (where c refers to the neutral carba analog) to five specific H3K4me3 readers, the PHD domains of JARID1A, BPTF, TAF3, the Tudor domains of the Royal Family of SGF29 and JMJD2A was studied. All the readers have aromatic cages for specific trimethylated lysine binding, but all have a different architecture of the typical motif. It was shown, that H3K4me3, which is positively charged binds 2-33-fold stronger, than the neutral H3c4me3 to readers that contain a Trp residue in their aromatic cage [55]. Interestingly, the association of H3K4me3 is more favorable enthalpically, but less favorable entropically, compared to the association of H3c4me3, in the same aromatic cages [55]. The two histone peptides bind with indistinguishable thermodynamics of associations to half-aromatic cages, indicating little to no contribution from cation-π interactions to the binding [55]. Aromatic cages containing tryptophan residues show stronger cation-π interactions binding to quaternary ammonium ions, if compared with aromatic cages containing phenylalanine and tyrosine residues [55]. 

ITC is also applicable for the thermodynamic analysis of large complexes of the epigenome. For example, binding of aprataxin and polynucleotide kinase like factor (APLF) to histone dimers and tetramers H2A-H2B and (H3-H4)2 was investigated in a study [141]. It was found that both histone systems bind the APLF reader protein with micromolar affinity, both are enthalpically favorable interactions, yet the binding of (H3-H4)2 is entropically unfavorable, which might explain the difference in their Kd values. ITC did not detect additional lower affinity binding modes, resulting in a stoichiometry of *n* = 1 in both cases [141].

Surface plasmon resonance (SPR) techniques are essential for high-throughput probing of biomacromolecular interactions. Kd is the main outcome of SPR which generally correlates well with the Kd from ITC measurements [119]. In contrast with ITC, the SPR measurements also provide reaction kinetic information of association and dissociation rates which can be useful in estimation of the kinetic stability of a drug-target complex [142]. Application of SPR yielded excellent comprehensive studies. For example, 125 types of modified histone (H1-H4) peptides with different PTMs were investigated [119] in combination with 8 histone reader proteins, resulting in one thousand pairs of interactions. It was discovered that KDM5A (also known as JARID1A, see in Category 5 of Section 1) interacts with H3K4me3 specifically [119]. KDM5A also interacts with human estrogen receptor and plays a role in osteogenesis. The study also showed that heterochromatin protein 1, important in the DNA repair after UV-induced damage, interacts with H3K9me3 [119]. Another study [143] used 204 proteins from either the Royal Family, PHD, bromodomains or CW domains and subjected them to SPR investigations with three specific histone modifications (H3K4me3, H4K5acK8ac, H3S10ph). The results were confirmed by ITC. It was found that the Tudor domain of echinoderm microtubule-associated protein-like 1 (EML1) binds to H3K3me3 with a greater affinity than to H3K36me3 [143]. EML1 is associated with Usher syndromes, which is a disease that eventually progresses in the whole brain. SPR was also used [50] to investigate the dependence of histone H3 binding to WDR5 on the methylation state of H3K4. The peptides were immobilized in the analyses, wild-type and mutant proteins were also assessed. H3K4me2 shows the strongest binding to WDR5, mono- and tri-methylated peptides bind seven and eight-fold weaker. Interestingly, H3K4me2 has both the smallest association and dissociation rate, if compared with the otherwise methylated H3K4 peptides. The small k_on_ rate indicates a slower approach to equilibrium and k_off_ corresponds to a sluggish decay of bound peptide signal when dissociating. This might be explained by an extended intracomplex interaction formed by H3K4me2 if compared with other modified H3K4 peptides, involving a hydrogen-bonding network between the ligand, water molecules and a backbone amino acid residue of the target [50]. For all histones, a micromolar Kd was measured, indicating a relatively weak binding affinity, compared to for example, strong, small molecule inhibitors [50].

Fluorescence spectroscopy is a versatile and sensitive method, used for investigations of a wide range of interactions including histone binding [95,130] and nucleic acid modifying enzymes [144]. Fluorescence spectroscopic determination of Kd of histone–reader complexes often completes PTM studies to affirm the findings of ITC measurements [95,130]. 

The methyl-CpG-binding domain protein 3 (MBD3), a DNA methylation reader was investigated in living cells, under hypoxia and decitabine treatment [145] by fluorescence correlation spectroscopy. The changes in the environment alter the fluorescence of the reporter (green fluorescent protein), which is suitable for detecting conformational changes at the timescale of milliseconds [144]. By monitoring the dynamics of the MBD3 protein, a fast diffusion in the nucleosome was observed, showing a form of demethylation that is independent of DNA replication [145]. Fluorescence correlation spectroscopy also contributed to identifying hypoxia sensitive cells and the real time follow-up of demethylation, which occurs in context with the hypoxia [145].

Fluorescence resonance energy transfer (FRET) assays are also popular tools to study, e.g., estrogen receptors and their coactivators [146], DNA bending by charge variant bZIP proteins [147], and transcription factor binding kinetics to nucleosomes and DNA [148]. 

Besides determination of atomic resolution structures (Section 2) NMR techniques are important in ligand-based binding assays and hit generation [149]. Such measurements are based on the spectral differences of hits and non-binding ligands. Target immobilized NMR screening can use the same target sample for a considerable number of ligands [149]. A study [141] of APLF nicely shows how NMR can supply both binding and structural information in the same experiment. APLF is a DNA repair factor with histone chaperone activity of its acidic domain, which uses two aromatic side chain anchors towards histones H2A and H2B. NMR titration experiments with stoichiometric addition of reagent solution (APLF acidic domain) to H2A and H2B were performed and peak intensity ratios and residue-specific chemical shift perturbations were collected from N-TROSY spectra. 2D NMR line shape analyses of the data resulted in Kd and also kinetic (k_off_) values, and a distinction was possible after fixing the initial values, to investigate the formation of a secondary complex, a secondary binding event with lower affinity [141]. With the calculated chemical shift perturbations, key residues of the complexes were also identified, allowing assignation of the structural origin of the binding affinities.

Inhibition assays are common, fast methods for estimation of binding affinities of inhibitors to enzymes. Such assays often produce the half maximal inhibitory concentration (IC_50_) values which can be related to the thermodynamic inhibition constant (Ki) [150]. Thus, IC_50_ is system-dependent, and not directly applicable instead of Ki (Kd). It can be applied for fast comparison or screening of a series of ligands according to their inhibitory effect on the same target enzyme. However, this level of information is often sufficient for further investigations or drawing conclusions. For example, a small molecule inhibitor was tested in vitro in mantle cell lymphoma models [151]. Protein arginine methyltransferase 5 (PRMT5) is overexpressed in patient samples with mantle cell lymphoma. Small nuclear ribonucleoprotein Sm D3 (SmD3), is a protein involved in RNA splicing, and is methylated by PRMT5. Observing the methylation of SmD3 with biochemical assays, PRMT5 enzyme activity was measured. The study [151] suggests that observed antiproliferative effects were a direct result of PRMT5 inhibition by the small molecular inhibitor that showed an IC_50_ value of 22 nM. In general, lysine methylation is a better studied area in cancer pathogenesis than arginine methylation; however, this study [151] offers an insight into arginine methylation in cancer, and represents a validated chemical probe for further studying. 

In many epigenetic studies, a combination of various experimental methods is applied resulting in a more complete and reliable picture of the interactome. For example, a study combined FRET and ITC to measure selective inhibition of HDAC isoforms [152]. Results of a set of experimental methods (in vitro binding assays, NMR binding, fluorescence titration assays, ITC, expression analysis and chromatin immunoprecipitation) provides a solid basis for further, structural investigations of, e.g., the effects of histone methylations [55,130] with theoretical approaches (Section 3).

#### 2.2.2. Biological Activity

There is a wide range of methodologies for measuring the activities of biomacromolecules and their ligand partners. Whereas binding affinity information (Section 2.2.1) is very important for molecular engineering, activity measurements provide high level tests of the new molecule. Thus, affinity and activity data complement each other, and in many studies, both types of measurements are present for the same system. In general, in vitro tests precede in vivo investigations, as the latter ones are rather costly, and therefore, they are applied mostly on a thoroughly screened, narrow set of compounds. 

##### In Vitro Activity

In a recent review [153], various in vitro methodologies including fluorescent, electrochemical, and surface-enhanced Raman spectroscopy-based assays were featured for investigations of histone PTMs and histone modifying enzymes. Fluorescent assay can measure e.g., the de-acetylation of peptides by HDAC enzyme [154]. The activity of HAT, adenovirus E1A-associated protein (p300) can be detected by monitoring coenzyme A formation in an electrochemical assay [155]. The activity of histone demethylase 1 enzyme can be also measured by detection of product concentrations in surface enhanced Raman spectroscopy-based assays [156]. In the next paragraphs, selected assays with some recent applications are also reviewed.

In vitro DNA methylation assay measures the enzymatic activity of DNMT3A protein (see Categories 3, 5 and 6 of Section 1). After methylation of DNA by DNMT3A, the isotopically labeled methyl groups can be detected. The assay was applied for the investigation of the effects of histone lysine residue methylation on DNMT3A enzyme autoactivation in an interesting study [157]. Relative enzyme activities were measured, in the presence and absence of histone peptides H3K4me0, H3K4me3, and the catalytic domain (CD) of the DNMT3A enzyme. In the presence of histone H3K4me3 the DNMT3A protein preferred an autoinhibitory conformation, in which its ADD (Section 1) does not bind to the histone tail. The binding of histone H3K4me0 to the ADD domain of DNMT3A induced a conformational switch favoring the active form of the enzyme [157], allowing the formation of the DNA-CD interaction (Figure 7).

An activity assay can also be applied to uncover the mechanism of the enzyme involved. The histone-lysine N-methyltransferase SUV39H1, uses H3K9me1 as a substrate to create H3K9me3 [158]. This methylation leads to a condensed state of the chromatin (see Category 5 of Section 1), and the exact mechanism that lies beneath this phenomenon was investigated in the study [158]. The enzyme has a CD and an N-terminal end, which are also important for its function. First, in its free form the enzyme samples chromatin through its CD. Then recognition of H3K9me3 allosterically activates a chromatin binding motif to anchor the enzyme with the likely involvement of its N-terminal segment, promoting H3K9 methylation. An active site mutation resulted in the inactivity of SUV39H1. Disabling the active site of the CD of the enzyme led to a disruption of promotion of H3K9 methylation through the N-terminal segment. Accordingly, the enzyme mutant lacking the N-terminal end showed lower activity [158]. The addition of H3K9me3 peptide strongly enhanced enzyme activity, by a tenfold increase in maximum velocity, and a fourfold reduction of the substrate concentration required to reach half-maximal rate [158]. This finding underlined the allosteric activation by H3K9me3. The promotion of H3K9me3 was called spreading in the study [158], and it resulted in a spatial closure of the nucleosomes. In this study [158], causal relationship was established with respect to how a PTM is transferred as a function.

Colorimetric assays, sulforhodamine B (SRB), mitochondrial metabolic activity (MTT) and crystal violet (CV) are used to determine cell viability [159]. The SRB, MTT, CV and LDH assays were applied to measure the cytotoxicity of apicidin on human pancreatic cancer cell lines, Capan-1 and Panc-1 [159]. Apicidin is a HDAC inhibitor. HDACs catalyze the deacetylation of primarily lysine residues at the N-terminal tails of histones [159]. The HDAC enzyme family takes part in chromatin remodeling and modification of gene expression (see in Categories 1, 2 and 5 of Section 1), and in the consequent pathogenesis of various malignant diseases. Pancreatic cancer cell lines were cultured, grown and plated before the experiments. Apicidin, a HDAC inhibitor was used at different concentrations for incubation with cell lines. The effects of short duration and longer exposure were investigated, and non-treated cells were used as control. EC50 values of apicidin were measured by these assays, and a dose dependent cytotoxicity was detected after 24 h treatment. An increase in cytotoxicity and decrease in cell viability was observed after treatment with 100 nM or higher doses of apicidin [159]. Moreover, in pancreatic cancer cell lines, apicidin showed an initial antiproliferative effect before the onset of cytotoxicity [159].

MTT cell viability assay was used to measure the effectiveness of tamoxifen and anacardic acid on MCF-7 and T47D breast cancer cell lines [160]. In an effort to unravel disease pathomechanism, and find possible therapeutic targets, epigenetic-related markers were screened, including oxidative forms of DNA-methylation, histone modification and methyl-binding domains to identify H4K12ac and H3K27ac as potential epigenetic therapeutic targets [160]. Anacardic acid, a HAT (see Section 1) inhibitor reduces the levels of acetylated H4K12ac and H3K27ac [160]. Then, it was combined with tamoxifen, a widely used agent in the treatment of breast cancer. The cell lines were cultured and incubated, as a pre-treatment for MTT cell viability assays [160]. The assays were then performed by addition of anacardic acid and tamoxifen, and a second incubation time was introduced before analysis. A solvent, not containing any cells were used as background [160]. The combination of tamoxifen and anacardic acid resulted in a marked inhibition in cancer cell viability with an additional loss of FRET efficiency between ERα and histone acetylation marks. Such combined epigenetic and hormone receptor mediated pathomechanism of breast cancer results raises the possibility of a combined treatment targeting multiple pathways of the disease [160].

Similarly, cell viability was measured by MTT assay, after treatment with a HDAC inhibitor, on pediatric embryonal cell lines [161]. HDACs are often used to treat various malignant diseases (see in Categories 1, 2 and 5 of Section 1), screening of compound libraries for HDAC activity is an emerging area of drug development [161]. A potent novel agent, HKI46F08 was tested on pediatric neuroblastoma and medulloblastoma cancer cell lines [161]. Its EC50 value was in the range of 0.1–4 µmol/L. Furthermore, HKI46F08 induced cell differentiation and apoptosis, overall being a promising agent in pediatric malignant diseases [161].

A cell proliferation assay, based on the measurement of optical density was used to study the effects of a small molecule inhibitor on gastric cancer [162] and gastrointestinal stromal tumor cell lines [163]. C646, a HAT inhibitor (see Section 1), inhibits the enzymatic activities of p300 and CREB binding protein (CBP). These HATs control the acetylation of histone H3, their inhibition exerts antineoplastic effects on various cancer cell lines [162,163]. In two studies [162,163], it was tested whether their inhibition provides beneficial effects against two types of malignant diseases of the gastrointestinal tract. In the first study [162], CBP and p300 enzymes were overexpressed in five gastric cancer cell lines. The control was a normal gastric cell line. C646 was also added to all of the cell lines. Optical densities were measured after incubation, and cell proliferation was calculated by dividing the optical density of the active set with the optical density of the control set. Higher doses (> 10 µmol/L) of C646 resulted in a stronger inhibition of cell proliferation on gastric cancer cell lines, than on normal gastric epithelial cells. In addition, it increased the number of apoptotic cells in gastric cancer cell lines and reduced migration and invasion potential [162]. C646 treatment reduced the acetylation of histone H3 in both gastric carcinoma cells and normal gastric epithelial cells. In the second study [163], the same protocol was repeated for gastrointestinal stromal tumor cell lines, with the introduction of an add-on treatment with imatinib. It was found, that alone 15 µmol/L C646 caused a marked decrease in cell proliferation of gastrointestinal stromal tumor cell lines, which result was further improved when combined with 500 nmol/L imatinib [163].

##### In Vivo Activity

In vivo activity tests cover investigations on living animals with various approaches. For example, in vivo enzyme activity tests can be performed by magnetic resonance (MRI)-based methods, microdialysis and fluorescence imaging [164] among others. In vivo MRI-based methods can apply microinjection of contrast agents cleaved by a specific enzyme to map the gene expression of transgenic animals, and improve our knowledge of mRNA expression, inheritance patterns and plasmid gene expression [165]. A selection of further in vivo studies on the epigenome is detailed in the forthcoming paragraphs.

In vivo chromatin immunoprecipitation (ChiP) assays are frequently used in recent works. ChiP is performed by cross-linking DNA and associated proteins, then fragmented DNA segments associated with proteins are extracted from the debris by protein-specific antibodies. The DNA segments are then purified, and their sequences are determined. With this approach, locations in the genome associated with specific histone PTMs can be screened. For example, ChiP assays identified that AIRE forms complexes with small fractions of H3K4me0, but not with H3K4me3. Furthermore, the specific promoter regions of DNA interacting with AIRE were found [130]. Promoter regions, where mostly H3K4me0 is expressed, like the insulin promoter region interact with AIRE. At the same time, regions that lack H3K4me0 but rich in H3K4me3, like the glyceraldehyde-3-phosphate dehydrogenase promoter region does not interact with AIRE [130].

ChiP-sequencing (ChiP-seq) combines ChiP assays with parallel DNA sequencing, similarly, to map DNA binding sites of proteins. After the ChiP assay, all DNA fragment sequences are determined in parallel, for a genome-wide analysis. In a ChiP-sequencing study [166], nuclei were extracted from midbrain dopamine producing neurons (mDA) of adult mice to create ChiP-seq libraries. The presence of repressive and permissive histone PTMs, H3K27me3, H3K9me3 and H3K4me3 around transcription start sites were screened to gain a picture on how the equilibrium state of the chromatin correlates with gene expression rates [166]. Occurrence of H3K4me3 was associated with high expression if compared with the total average gene expression level even when co-occurring with repressive modifications. The distribution of other histone modifications also correlated with gene expression as chromatin regions rich in H3K27me3 and H3K9me3 corresponded to lower than average gene expression. The simultaneous presence of H3K27me3 and H3K9me3 were associated with terminal repression of the gene expression. This chromatin equilibria regulation is maintained during transition from neuronal progenitor cells (NPCs) to mDA. The already H3K27me3 enriched genes not only maintain their repressed states in equilibrium, but even gain additional H3K9me3 marks upon transition from NPCs to mDA [166]. 

The xenograft tests are applied in tumor growth studies in vivo, where cancer cell lines are transferred into animals, then control and treated groups are formed to study the effect of a drug on tumor size. For example, a drug named corin was tested in a melanoma xenograft mice model [54] for tumor growth modifying effects [54] through targeting epigenetic pathways. A therapeutic target of special interest, that contains a HDAC enzyme (see in Categories 1, 2 and 5 of Section 1), namely CoREST complex (see in Category 5 of Section 1), consisting of REST co-repressor 1 protein (CoREST), HDAC1 or HDAC2 and LSD 1 enzymes. Corin was tested as a dual action LSD1 (see in Category 5 of Section 1)/HDAC inhibitor targeting the CoREST complex. Corin showed metabolic stability and proved to be well-tolerable in mice. Mice were divided into vehicle and corin treated groups. Following euthanasia of the animals, tumors were collected and measured. Corin showed a marked reducing effect on tumor growth compared to vehicle. Tumor cells extracted from these mice showed an elevated H3K9ac acetylation and H3K4me2 dimethylation in corin-treated mice, compared to vehicle-administered mice [54]. This observation correlates well with the HDAC and demethylase inhibitor functions of the drug, corin.

In another xenograft study [160], sixty-day releasing 17ß-estradiol pellets were subcutaneously inserted into the shoulders of eleven to fifteen weeks-old female mice [160]. MCF7 cells (See Section 2.2.2) were also subcutaneously inoculated into the animals. After the tumor reached a certain size, the mice were divided into groups, and the treatment was initiated. The control group was injected with solvent, treatment groups were injected with tamoxifen and anacardic acid. Tumor xenograft volumes were then measured. The tumor growth was reduced compared to control groups [160], this is in good agreement with the in vitro results of the same study [160], detailed in Section 2.2.2. The combined treatment with tamoxifen and anacardic acid inhibited ER-regulated gene transcription. Anacardic acid alone showed a reduction in H4K12ac occupancy near growth regulation by estrogen in breast cancer 1 (GREB1) transcription starting site, tamoxifen alone did not exhibit this effect [160].

In tumorigenesis studies, the growth of a tumor is also induced in animals, like in xenograft studies, but with the intention to study the pathomechanism of a certain type of tumor, or the pathologic pathway induced by an agent. The contribution of epigenetic changes to the carcinogenicity of potassium dichromate (further referred to as CrVI) was investigated in a study [167]. CrVI is a known genotoxic carcinogen. CrVI-transformed cells from human lung cancer tissues and CrVI-exposed human bronchial epithelial cell lines were injected into female nude mice [167]. Chronic CrVI exposure increased histone-lysine methyltransferase expression, and consequently repressive H3 methylation marks, playing a causal role in the carcinogenicity of CrVI. Gene knockdown or pharmacological inhibition of the histone-lysine methyltransferase diminished this effect [167].

Among invasive sampling methods microdialysis is often used for continuous measurements of unbound analyte concentrations of the extracellular fluid. For example, dopamine levels were measured in mice brain after alcohol administration, to investigate the effect of alcohol on histone acetylation patterns [168]. Microdialysis was performed by inserting a dialysis probe into the brain tissue of mice above the nucleus accumbens. The probe was perfused with artificial cerebrospinal fluid at a constant rate. Baseline samples were taken to measure the baseline neurotransmitter levels. Animals were injected either with ethanol or saline and samples were collected every 20 min through the microdialysis probe [168]. It was found [168], that ethanol administration provokes similar prolonged dopamine response in both adolescent and adult rats, but basal dopamine levels were higher in ethanol-treated adolescent rats, than in similarly treated adult rats [168]. Finally, ethanol administration changed the histone H3 and H4 acetylation of adolescent rats in the nucleus accumbens, striatum and frontal cortex. The study [168] concluded that epigenetic changes might contribute to the increased vulnerability of adolescent rats to alcohol addiction. 

## 3. Theoretical Calculations of Molecular Structure and Binding Affinity

The previous paragraphs of Section 2.1.2 on experimental methods highlighted the most important challenges of determination of molecular structures in the epigenome. It was explained how the high number of variations meet large biomolecular system sizes, resulting in an extraordinary problem to be tackled. Such complexity of the epigenetic interactome clearly shows the limits of experimental approaches. In this situation, the use of theoretical approaches is inevitable to enhance the production of new structural information and also to predict the strength of corresponding molecular interactions. Although some of the theoretical methods require an advanced computational infrastructure, the cost of such facilities is still moderate if compared with that of experimental studies. Moreover, due to the general need and spread of information technologies in all fields of society, their development obviously shows an increasing trend. This has a positive feedback effect on the scientific applications often resulting in higher benefit-cost ratios in both the software and the hardware components of these technologies. Beyond a complementary use of theoretical approaches, the forthcoming paragraphs also highlight their advances over physical experiments in problematic cases where measurements are not available or reached their natural limits. The survey of the forthcoming paragraphs includes examples at various levels of theory.

Knowledge-based approaches are trained on sets of experimental molecular structures and their physico-chemical background is restricted to basic principles only. They often depend on comprehensive databases and internet services such as the Protein Databank [87], the Basic Local Alignment Search Tool (BLAST, [169]) or FASTA [170]. Knowledge-based methods provide fast results, often implemented in on-line servers, and do not require extensive computational infrastructure. However, the reliability of their results is limited by the training data set and the state-of-art of the databases and servers working in the background. Molecular mechanics (MM) methods are generally published as standalone tools based on more sophisticated physical chemistry, but still working by the laws of classical physics [171]. MM methods allow not only local search and fast optimization of the structures, but also extensive conformational sampling and global search in molecular dynamics (MD), Monte-Carlo or genetic algorithms. Such features are of particular importance during investigations of structural (Section 3.1) and energetical (Section 3.2) properties of molecular interactions, also accounting for interface flexibility, or predicting protein side chain conformations, and so forth [172,173,174]. 

### 3.1. Molecular Structure

There are two main goals of theoretical methods on structural calculations. They produce either an atomic-resolution structure of a single macromolecule, such as a protein target for drug design, or a complex structure of two or more partners involved in protein–ligand, protein–protein, protein–DNA or other interactions. Beyond production of such static structures (snapshots), recent molecular dynamics investigations often produce a series of molecular geometries presenting the evolution of the systems. This feature is of particular importance for the exploration of induced effects and drug binding mechanism. Accordingly, the next paragraphs will feature selected results of static (Section 3.1.1) and dynamic (Section 3.1.2) methods, as well.

#### 3.1.1. Static Methods

Among knowledge-based approaches, homology modeling is a primary structure prediction method of proteins and their complexes. It is a quick technique based on the assumption that similar sequences fold into similar structures. The technique requires access to on-line databases holding protein sequences [175], the above-mentioned sequence comparison algorithms (BLAST, FASTA) for selection of a template protein available in the Protein Databank. An acceptable homology model requires a large sequence identity between the modeled (target) and template structures [176]. The number of known protein sequences is higher than that of determined protein structures [177]. As protein structure is primary information for target-based drug design, homology modeling is often involved in such projects with epigenetic targets [52,178,179,180,181,182,183,184,185,186,187,188,189,190,191,192]. The homology-modeled protein targets can be used in virtual screening of chemical libraries to find lead molecules. For example, in a study [188], a 3D structure of KDM5A (see in Category 5 of Introduction and Section 2.2.1) jumonji domain was built by homology modeling. For building KDM5A jumonji domain the program MODELLER [193] was used, with four templates, which either contain a jumonji domain or a similar structure. Template proteins lysine specific demethylase 4C jumonji domain [194] and 2-oxoglutarate oxygenase [194] both had a ca. 40% amino acid sequence identity to the jumonji domain of KDM5A. Five models were generated from each template, and then the models were subjected to different MM minimization steps, to select the most appropriate model for the screening process. Then molecular docking-based virtual screening was performed on compound libraries, and structure activity relationship analysis was carried out, to identify novel potent inhibitors of the enzyme. KDM5A functions as a transcriptional repressor (see in Category 5 of Introduction), through the demethylation of H3K4me3 [188]. Dysregulation of KDM5A is involved in the pathomechanism of various human malignant diseases, such as breast cancer and acute myeloid leukemia [77,78,188]. In addition, it was shown, that KDM5A is involved in the drug resistance of anti-cancer drugs [188]. The hit compound identified in this study [188] showed an in vitro IC_50_ value of 0.22 µM on the KDM5A enzyme, a promising starting point for further investigations.

Besides homology modeling of single targets, combination of existing structures can also lead to the solution of target–ligand complexes. For example, the complex of AIRE PHD finger and histone peptide H3K4me0 was modeled [130] by superimposing the histone complex of NURF BPTF PHD finger (see in Section 1, [80]) and the apo structure of AIRE PHD finger [195] by the program Lsqman [196]. Linear disordered C and N terminal parts of AIRE PHD finger and the NURF BPTF PHD finger were then removed, and the remaining AIRE PHD finger was blocked by acetyl and amide groups. The system was refined by MM energy minimizations with GROMACS [197]. One year later, the solution NMR structure of the complex was also captured [95]. Comparison of the modeled and experimental structures (Figure 8) show that the binding mode of the H3 peptide ligand perfectly matches in the two structures including the three β-strands in antiparallel organization and important side-chain interactions between the histone peptide residues H3R2, H3K4, H3T6 and backbone residues of the AIRE PHD finger C310, L308 and G306, respectively. Furthermore, backbone carbonyl oxygen atoms of residues P331-G333 anchored the N-terminal end of the histone peptide tail through intermolecular hydrogen bonds. Hydrophobic interactions occurred between the methyl group of H3A1 and the pyrrolidine ring of P331, and the methylene groups of H3K4 and L308. Finally, two salt bridges were formed between the side chains of H3R2 and D312 and H3K4 and D297 [130]. 

Calculation of structures of target–ligand complexes is an important goal of MM- or knowledge-based computational docking approaches, as well. Initially, fast docking programs were introduced to select small-molecule drug candidates during rapid screening of ligand libraries (see previous paragraph on KDMA5 target for an example). The technique is also useful for predictions on the target side during investigation of the effects of amino acid mutations involved in ligand binding [198,199,200]. Beyond small molecule drug candidates, there is a rising interest in peptides as ligands. However, flexibility of both the target and the peptide ligand [201,202,203,204] is challenging for fast docking methods [205] which often neglect conformation flexibility to reduce computational cost [100]. 

Numerous servers have become available based on fast docking approaches [202,204,205,206,207,208,209,210,211,212,213,214]. Some of these are designated for peptide ligands [207,209,210,212,214,215]. For example, the FlexPepDock [207] server was used to dock H3K4me3 peptide on transcription factor 19 (TCF19) PHD [216]. The homology model of TCF19 PHD was created on the basis of the PHD finger of the Jumonji/ARID domain containing protein 1A, this template has a 50% amino acid sequence identity with the TCF19 PHD. The molecular mechanism behind the function of TCF19 was explored, and it was found, that TCF19 PHD selectively interacts with histone H3K4me3 mark, and recruits the co-repressor complex NuRD (Section 1), to regulate gluconeogenic gene expression in HepG2 cells [216]. 

While the field of fast docking methods is rapidly developing, the above problems of peptide ligands have not been solved yet. In particular, the docking of histone peptides to their targets (readers) is still problematic and rarely addressed. To address this problem, a fragment blind docking [99] strategy was introduced and tested for docking of 7–13 amino acid long histone N-terminal tail peptides. Selected epigenetic proteins were targeted (references to the Sections indicate the places of further explanations on each protein) including AIRE (Section 2.1.2), ATRX-ADD (Section 1), DNMT3L (Section 1), KAT (Section 1) and Set domain containing protein (Section 2.1.2). The strategy applied Wrap’n’Shake [100], a blind docking method wrapping the target surfaces with a monolayer of copies of dipeptide ligand fragments. Then the full peptide ligands were reconstructed by linking the fragments. With this strategy, good agreements were achieved with experimental structures for the N-terminal part of histone peptides. For example, the N-terminal ARTK peptide of H3 showed a low root mean squared deviation (RMSD of 1.3 Å) from the experimental conformation [99]. Notably, the N-terminal segment has primary role in histone interactions and structure determination of H3 complexes is often restricted to this region (see also Section 2.1.2). 

The results of virtual docking screens are often piped into in vitro assays for final selection of the top candidates. For example, the discovery of novel DNMT3A (Section 1) inhibitors was aided by structure-based virtual screening and in vitro DNMT3A inhibition assays [217]. DNMT3A is responsible for the methylation of cytosine at C5 position [217]. DNA hypermerthylation of specific genes contributes to cancer initiation and progression [217]. Specifically, DNMT3A mutations are associated with haematological malignancies [217]. In a study, over 77,000 commercially available molecules were subjected to virtual screening via molecular docking for an X-ray crystallographic DNMT3A structure [217]. The molecules were docked onto the S-adenosyl-l-homocysteine site of the enzyme, with Glide [218] DOCK [219] docking programs. The top ranked molecules were evaluated by AMBER [220] scoring, and the remaining 1000 molecules were merged into one file for cluster analysis [217]. 107 molecules were then evaluated by in vitro DNMT3A inhibition assays, and two compounds displayed significant in vitro inhibitory activity with IC_50_ values of ca. 40 μM [217].

Similarly, docking-based virtual screening was performed on SPECS database, to identify novel non-nucleoside DNMT1 inhibitor compounds [221]. DNMT1 is the most abundant among DNMTs, yet non-nucleoside inhibitors are lacking against DNMT1 [221]. Non-nucleoside inhibitors do not show as many side effects, as nucleoside analogs, unfortunately they also show lower potencies than nucleoside analogs [221]. An X-ray crystallographic structure was used, and similarly, its S-adenosyl-l-homocysteine binding site was searched, and after discarding ligands with unfavorable physicochemical properties, over 110,000 compounds were screened against DNMT1 [221]. After scoring by Glide [218], 51 compounds remained for further evaluation in biochemical assays. A compound, DC_05 showed remarkable selectivity for DNMT1 isoform, with an IC_50_ value of 10.3 µM [221]. Afterwards, similarity-based analog searching was performed, with DC_05 as a lead compound, and two even more potent agents were found, DC_501 and DC_517 with IC_50_ values of 2.5 and 1.7 µM, respectively [221].

#### 3.1.2. Dynamic Methods

Besides production of a static snapshot of a single conformation (Section 3.1.1), uncovering interaction dynamics is another key to epigenetic regulation. MD can produce a time series of conformations of proteins or any molecular assemblies. It can be used to check the long-term stability of folding and complexes of large molecules even on a ms scale. Explicit solvent MD simulations allow very precise calculations accurately modeling real solutions.

MD simulations can be used to study the effect of PTMs on the stability of histone-reader complexes. For example, the effect of methylation of H3K4 on its binding strength to AIRE PHD finger was investigated using GROMACS [222] software package with GROMOS96 [223] force field [130]. Instead of the H3K4me0 peptide (Figure 8) H3K4me3 with trimethylated lysine side chains was applied and neutralizing counter ions and 7800 explicit single point charge waters were used in rectangular simulation boxes [130]. Short MD simulations were performed on four different complexes including AIRE PHD finger in complex with four possible PTM variants of the K4 amino acid, respectively. It was found that the H3K4me0 variant had stable complex with the first β-strand of AIRE PHD finger, creating an antiparallel β-sheet, stabilized by the interactions listed in Figure 8 of Section 3.1.1 [130]. Two salt bridges were formed between histone H3R2 and D312 and H3K4 and D297 residues [130], which is crucial for complex stability. Increasing the number of methyl groups (H3K4me, H3K4me2) destabilized the complex. In the case of H3K4me3, the complex could not be stabilized, and the two partners quickly dissociated in the MD runs. The bulky trimethylamino group of H3K4me3 could not participate in the above salt bridge hindered by several possible clashes with the target surface [130].

The effects of interaction networks of water molecules on complex stability can be also investigated by MD. An example of the ternary complex of DAXX protein and histones H3.3 and H4 was described (Section 2.1.2) in detail previously. In another example, MD simulations with GROMACS and MobyWat [127] were applied for the calculation of interfacial hydration network of ATRX-ADD protein in complex with histone H3 tail, trimethylated at H3K9me3. The binding of ATRX-ADD (Section 1) to histone H3 tail is promoted by H3K9me3 PTM mark and inhibited by H3K4me3. After mutation of the trimethyllysine binding pocket of ATRX-ADD it cannot bind to histone H3K9me3, and pericentromeric heterochromatin, leading to apoptosis in neuroprogenitor cells and mental retardation syndrome [81]. The relatively large histone H3 tail interacts via a shallow binding interface, where its arginine and lysine side chains are open to interact with water molecules from the bulk solvent. MobyWat showed that 12.5% of the water molecules of the system has low mobility, and is involved in a static sub-network [127]. The formation of such static networks are essential for complex stability anchoring the N-terminal tail of the histone to the target molecule, and can also shield the target-ligand H-bonds from solute attacks [127].

Two bromodomains, CBP (Section 2.2.2) and bromodomain adjacent to zinc finger binding domain 2B (BAZ2B) [224] were also analyzed in MD simulations, and conserved water molecules were studied at the bottom of the acetyl-lysing binding site of the bromodomains. The movement of the ZA loop of the binding site of the bromodomains has an influence on the presence of conserved water molecules in the binding site. These water molecules are connected by hydrogen bonds and were all either present or absent along the simulation [224]. Co-solvents, DMSO and (m)ethanol were added to the system, and similar results were achieved, with available crystallographic conserved bromodomains with the same co-solvents [224]. In their most populated binding modes, the co-solvents accepted a hydrogen bond from the same asparagine residue that is involved in the binding of acetyl-lysine [224]. Upon reaching more buried binding modes, the co-solvents displaced the same structured water molecules during the MD simulations. It was concluded in the study [224], that during ligand design, only the structured water molecules, that do not exchange with bulk solvent should be kept in crystal structures, during docking runs [224], and the identified water molecules, displaced by (m)ethanol co-solvents, might be targeted by hydrophilic moieties of the ligand [224].

MD simulations also uncovered the structural background of substrate selectivity of lysine specific demethylase 4A (KDM4A or JMJD2A) [225]. JMJD2A is a histone demethylase, specific for di- and trimethylated H3K9 and H3K36. The expression of JMJD2A is increased in prostate cancer [78]. MD simulations with mono-, di-, and trimethylated H3K9 peptides and JMJD2A were performed. The JMJD2A enzyme has a Fe^2+^ ion in its active site, to which a water molecule is coordinated, which does not form any hydrogen bonds with its surrounding atoms [225]. This water molecule stayed coordinated to Fe^2+^ throughout the whole simulation (20 ns). In all three cases of the PTMs, the water molecule was located always between the Fe(II) and the methylammonium moiety [225]. In the case of the mono- and dimethylated peptides, water molecules occupied the place of the missing methyl groups. These water molecules play an important role in ligand orientation within the binding pocket of JMJD2A, for example a water molecule, that stayed close to the methylammonium heads of the ligands through the simulation, formed hydrogen bonds with serine and glycine residues of the protein [225]. Apart from water molecules, from a structural point of view, the binding of H3K9me3 was found to be favorable, because of the symmetry of the ligand, which leads to an adequate orientation of the methyl groups. The preferable orientation of the methyllysine head in the case of H3K9me2 results from the restriction of angular motion by surrounding asparagine and glycine residues. If the dimethyllysine was rotated one of the methyl groups would overlap with the atoms of the surrounding asparagine and glycine residues of JMJD2A. The energy barrier observed between the three minima of the torsion states of H3K9me2 methyllysine head prevented the head from a circular motion [225]. Furthermore, the H3R8 formed intramolecular hydrogen bonds with H3K9me2 and H3K9me3, this interaction has a favorable energy contribution to the ΔG of the ligand [225]. 

### 3.2. Binding Affinity

The relevance and experimental methods of the measurement of binding affinity were introduced in Section 2.2.1. The large number of molecular interactions in the epigenome (Section 2.1.2) necessitated the development of theoretical approaches for fast generation of binding affinity data. There are statistical and end-point methods [226] available for calculation of binding thermodynamics, mostly ΔG. The development of such structure-based approaches is a hot field of research due to their central importance in rational drug design.

#### 3.2.1. Statistical Methods 

The first group of methods uses sampling of a statistical ensemble of conformations of interacting molecules. MD (Section 3.1.2) is often used for production of such samples of billions of states of macromolecular systems also providing information for calculation of ΔS, as well [227]. Among the statistical methods, alchemical energy calculation methods involve the transformation of one ligand into another, or a non-interacting particle [226]. Pathway methods are somewhat computationally expensive and follow the whole path of the binding process a useful option in drug design [226]. 

Using alchemical atom-type mutations, thermodynamic integration technique was applied to calculate the ΔG of CpG DNA site with methyl-CpG binding domain protein 1 (MBD1), which binds to methylated sequences in DNA [228]. Via this binding event, MBD1 can influence transcription activity [228]. MD simulations uncovered the binding mechanism of MBD1 to a hemi-methylated DNA, where cytosine is only methylated on one DNA strand [228]. It was found that a hydrophobic path of MBD1 protein moves away from the demethylated cytosine, and this conformational change weakens the DNA-protein interaction [228]. During the binding process, bulk water enters the binding site at the interface, inducing the rearrangement of the hydrogen bond network and the loss of a crucial hydrogen bond, that would occur between methyl cytosine and a tyrosine residue of MBD1 [228]. On the other strand, due to these conformational changes, the hydrogens of the methyl group of the cytosine form hydrogen bonds with an arginine residue of MBD1 protein. In this way MD simulations contribute greatly to our knowledge on how methyl marks of the DNA is recognized in the epigenetic machinery [228]. The proposed mechanism was validated by experiments. The binding of MBD1 protein to fully methylated CpG DNA site is more favorable if compared with the unmethylated CpG DNA site [228].

In another alchemical paper, free energy perturbation was used [229] to quantify the interaction of methyl-lysine histone and its reader protein, lethal 3 malignant brain tumor like protein 1 (L3MBTL1). The calculated ΔG was validated by ITC measurements. It was found that an asparagine residue of L3MBTL1 protein acts as an anchor, and its mutation disables any measurable binding of histones [229]. Instead of histone peptides, probes were used in the calculations and experimental measurements, which were assumed to act similarly to histones. Interestingly, it was found that the addition of a methyl group to Nme0 (non-methylated amino moiety), or the removal of a methyl group from Nme3 results in an affinity gain (−4.73 kcal/mol and −2.11 kcal/mol, respectively) to the reader protein, while the addition of a methyl group to Nme1 (−0.3 kcal/mol) does not affect ΔG [229]. The atomic level background of this unusual phenomenon was investigated by MD simulations. It was found that Nme0 lacks all favorable van der Waals contributions; furthermore, its positive charge is shared between three hydrogen atoms, resulting in a considerable loss of electrostatic contribution to ΔG. Nme3 binding is penalized by steric repulsions, and positive energy contribution of non-polar terms, as well. Overall, Nme1 or 2 was concluded as a preferred PTM state of histone for binding to the L3MBTL1 protein [229]. 

Adaptive lambda square dynamics was applied [230], to calculate the impact of K14 acetylation on histone H3 conformation. It was found that H3K14ac results in a weaker interaction between the DNA and the histone H3 tail, and the acetyl mark enhances α-helix formation of the histone H3 tail. The favorable electrostatic interaction between H3K14ac and H3K18 leads to increased α-helix formation [230]. This results in a more compact tail conformation [230]. This compaction results in the unwrapping of the linker DNA from the nucleosome, and the exposure of the linker DNA [230], which enables DNA binding proteins (e.g., transcription factors, see Category 4 of Section 1), to bind to their target sequences [230]. 

The attach-pull-release method was used to calculate the binding free energy of seven small ligands to a bromodomain [231] (see also Section 3.1.2 on epigenetics relevance). During these investigations, the ligands were pulled off the bromodomain binding site, allowing its conformational relaxation. In this study [231], a conformational change in the bromodomain is revealed by MD simulations. In experimental apo crystal structures the bromodomain is in a closed state, which opens up in MD simulations after 20–60 ns run time [231]. In a loop, the two main chain asparagine residues undergo a transition of torsion angles, and other residues change only minimally. If a restraint on the torsion angle of one of the asparagine residues is applied, the conformational change does not occur [231]. The calculated ΔGs of the seven ligands were compared to experimental data from the literature. Additionally, various water models and ligand parameter set combinations were compared, both using the open and the closed states as the final apo state of the protein. Using the open state as the final apo state of the protein SPC/E [232] water model with GAFF force field [220,233] provided the best results compared to experimental data from the literature (RMSE 1.42 kcal/mol) [231]. The open conformation of the enzyme was found to be more favorable energetically [231]. As the transition of the apo protein to open state is thermodynamically favorable and promotes dissociation, keeping the protein in a closed state improves ΔGs [231]. This improvement in calculated binding free energies reduced the bias of computational results and led to a better agreement with experimental values. Interestingly, when the closed state of the protein was used as the final apo state the previously weakly performing TIP3P [234] and TIP4Pew [235] water models showed the best results (RMSE ranging from 1.14 to 1.61 kcal/mol).

#### 3.2.2. End-Point Methods

End-point energy calculations are based on the initial free ligand and target, and the final complex structures [226]. Due to the small number of conformations end-point methods are computationally efficient and fast [226]. Molecular Mechanics Generalized Born Surface Area (MM GBSA) [236,237,238,239,240,241,242,243,244,245] and Molecular Mechanics Poisson Boltzmann Surface Area (MM PBSA) [239,241,242,246,247,248,249,250] methods are commonly applied, single-trajectory approaches. The conformations of the interacting partners in their complex are assumed to represent the unbound partners [226] leading to several approximations [140,251].

The performances of MM PBSA and absolute alchemical binding free energy calculation methods were compared [252] using 22 different targets of epigenetic importance. Most of the calculations were performed on the members of the bromodomain and extraterminal (BET) family, including BRD2, 3, 4 and BRDT proteins. [253,254]. BET proteins regulate the expression of key oncogenes and anti-apoptotic proteins, making them a promising target in epigenetic drug design against malignant diseases, inflammation and viral infections [252,253,254]. Acetylation of lysine residues on the N-terminal tail of histone is associated with an open chromatin conformation and therefore transcriptional activation [254] (see also Category 5 in Section 1). Bromodomains are the readers of the acetylated lysine residues, the therapeutic approaches, and their small molecule inhibitors are reviewed in the literature [254]. In the study [252], abundant small molecular inhibitors of bromodomains were used in ΔG calculations. The calculated energies were compared to experimental ΔGs, and a thorough statistical analysis was performed. Absolute binding free energy calculations outperformed the MM PBSA approach for the investigated bromodomain complexes [252].

Scoring function of docking programs are often based on end-point ΔG calculations and applied in epigenetic drug design [218,255,256]. Scoring values usually show low correlation with experimental ΔGs [255], and therefore, they are mostly applied for distinction between ligand candidates, relative comparison of the members of a docked ligand library. This problem can be also addressed by consensus scoring [255]. In this case, if a hit is identified, the majority of scoring function methods have to rank it as the top ranks to get accepted. As scoring functions show inconsistent performance on different receptors, the careful selection of a scoring function is important for virtual screening [255]. Scoring functions GoldScore, ChemPLP, ASP and CDOCKER_ENERGY were applied [255] to support the hit discovery of sirtuin 2 (SIRT2) inhibitor. SIRT2 is a nicotinamide adenine dinucleotide-dependent deacetylase, that plays a role in the pathomechanism of various diseases, including cancer and neurodegenerative diseases [255]. It has a wide variety of substrates, including histone H3K18ac and H3K56ac, and H4K16ac [257]. Interestingly, within its display of substrates, there is a histone writer, histone methyltransferase PR-Set7 [257]. PR-Se7 specifically mono-methylates H4K20me0, but if the enzyme is deacetylated by SIRT2, its localization on the chromatin is altered, decreasing its ability to methylate H4K20me [257]. In the study [255], a SIRT2 inhibitor with a new scaffold was identified, and subjected to structure activity relationship analysis, and finally four compounds were developed with IC_50_ values less than 10µM against SIRT2.

Determination of components of ΔG is important in thermodynamic optimization of drug candidates (Section 2.2.1). In particular, the optimization of ΔH can increase drug efficiency [137,138]. To help this trend of drug design, end-point quantum mechanical (QM) approaches have been developed for structure-based calculation of ΔH [258,259]. While QM calculations offer the highest possible theoretical accuracy, they are expensive and demanding in computational time. Thus, the program Fragmenter was developed for the reduction of system size by extraction of hydrated interfaces from target-ligand complexes [136]. The complexes were subjected to semi-empirical QM calculations with PM7 parameterization, and the calculated ΔHs were correlated with available experimental values. Interestingly, the study [136] found a simple scaling factor for conversion between calculated to experimental binding enthalpies. Among other protein-peptide systems, the method was used to calculate the ΔH of the AIRE PHD finger–histone H3 peptide system featured in Figure 8 [136].

## 4. Conclusions

The present review featured current trends and selected methodologies of exploration of molecular structure, binding affinity and pharmacological activity in the epigenome. The design of efficient epigenetic drugs requires valid SARs and simultaneous development of all three fields. In recent decades, cryo-electron microscopy has opened a new avenue in the determination of molecular structure of nucleosome-sized assemblies. At the same time, high-throughput determination of atomic resolution structure of large biomolecules has not been implemented routinely. Thus, the application of alternative crystallographic, and theoretical approaches remains inevitable. Measurement of binding affinity is an important intermediate step during optimization of pharmacological activity. There is a wide range of techniques available at different levels of sophistication. Depending on the project, assays can be applied for fast screening of drug candidates or assessment of the effects of epigenetic modifications. On the other hand, sophisticated techniques like isothermal titration calorimetry help the optimization of the lead compounds providing detailed information on binding thermodynamics. Similarly, available theoretical binding affinity calculators provide fast and/or precise solutions using molecular structures as starting points. Molecular dynamics also helps to uncover binding mechanisms and supplies statistical amount of molecular conformations for energy calculations. In vivo activity experiments are essential for the final decision on further development of drug candidates and also provide a feedback for re-investigation of the epigenetic background of a disease. 

The reviewed methods have proved indispensable during the discovery of epigenetic drugs accepted for clinical use. Such U.S. Food and Drug Administration (FDA)-approved epigenetic modulating drugs include vorinostat, romidepsin, panobinostat, belinostat (HDAC inhibitors), azacitidine, decitabine (DNMT inhibitors), enasidenib and ivosidenibe (isocitrate dehidrogenase inhibitors) [260] and tazemetostat, a histone methyltransferase inhibitor [261,262]. Tazemetostat is a selective inhibitor of enhancer of zeste homolog 2 (EZH2), a histone methyltransferase, that trimethylates H3K27me3 [260,261,262,263,264]. Given that EZH2 is a transcriptional suppressor, histone methyltransferase, and transcriptional co-activator, it is involved in a wide variety of cellular processes, some of which are directly linked to cancer pathomechanisms [264], EZH2 is in the highlight of biotechnological and pharmaceutical companies. Tazemetostat is approved by FDA for the treatment of epithelioid sarcoma, malignant rhabdoid tumors, and integrase interactor 1 (INI1) negative tumors. Vorinostat [265,266,267,268,269], a HDAC inhibitor is used for the prevention of acute graft-versus-host disease, and the treatment of cutaneous T-cell lymphoma. The above-mentioned nine agents were approved between 2004 and 2018, highlighting the emerging role of epigenetics in current drug discovery and design. Further developments and spread of the above surveyed methods are essential but probably not sufficient criteria of future acceleration of the development of valid SARs and drug discovery in the epigenome. The invention of new computational technologies is necessary to handle the epigenetic SAR data universe and the improvement of their complementary applications with strong links to experiments is also inevitable. 

## Figures and Tables

**Figure 1 ijms-21-04134-f001:**
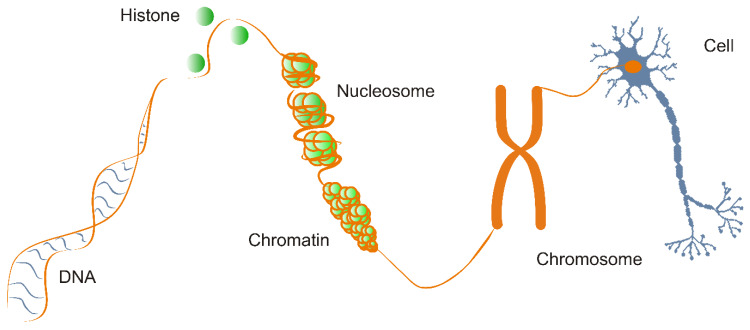
Structural background of the epigenome with a schematic illustration of major organizational levels. A neuron was selected to represent the cellular level as “precise epigenetic regulation may be critical for neuronal homeostasis” [21].

**Figure 2 ijms-21-04134-f002:**
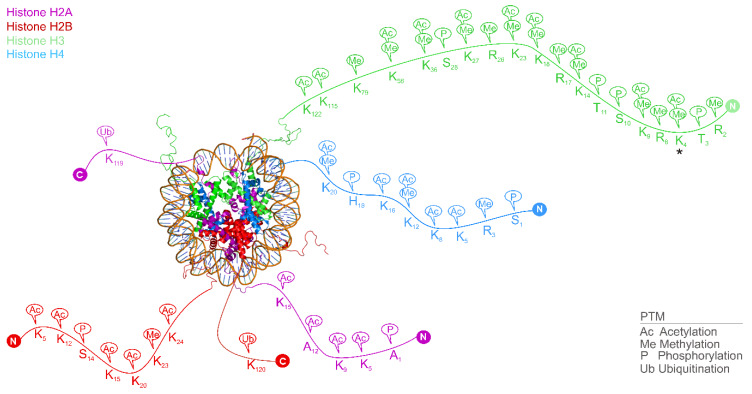
The three-dimensional structure of a nucleosome combined with a schematic representation of post-translational modifications (PTMs) on the histone tails assembled from recent articles [31,36,37,38,39,40,41,42,43,44,45,46]. The nucleosome structure was rendered in top view by PyMol [47] using PDB structure 1kx5 [48]. Histone proteins are shown in cartoon representation as wrapped by the DNA double helix. The N- and C-terminal tails of histone proteins pass through the cylinder of the supercoiled DNA and are available for reader proteins recognizing the PTMs, key components of the histone code system.

**Figure 3 ijms-21-04134-f003:**
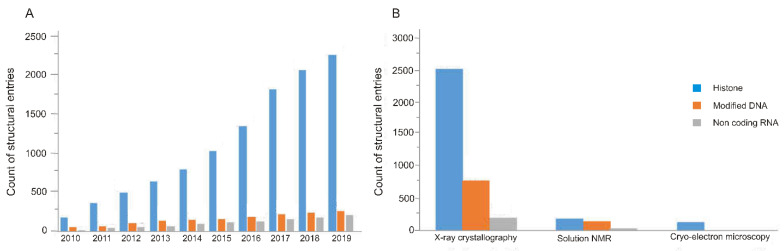
Trends of experimental structure determination of representative macromolecules of the epigenome. (**A**) The count of structural entries in the Protein Databank (PDB) per year (cumulative plot). (**B**) Distribution of entries of 2019 in (**A**) grouped by the main experimental techniques. The plots are based on a search in the PDB using key words ‘histone’, ‘modified DNA’ and ‘non-coding RNA’, which also involves ’siRNA’, ’miRNA’, ‘lncRNA’ were used in PDB. Accession date: 6 April 2020.

**Figure 4 ijms-21-04134-f004:**
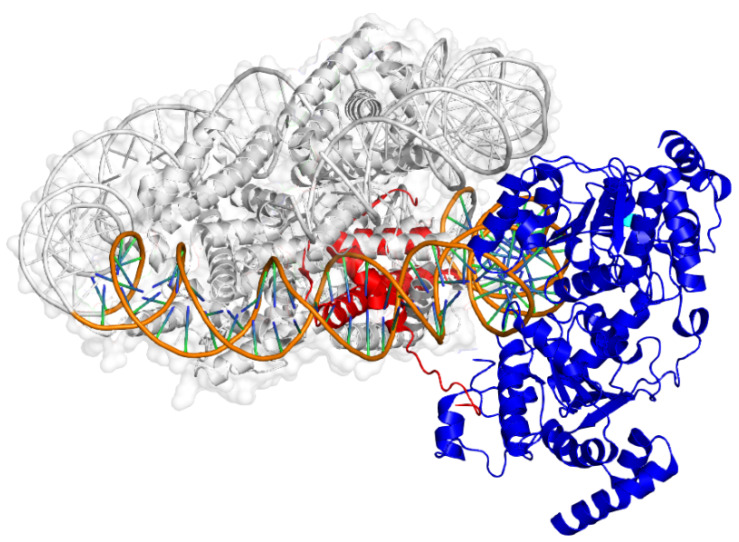
The ternary complex of STH1/NPS1 nuclear protein (blue cartoon), a histone reader, the DNA (orange cartoon), wrapped around the nucleosome and histone H4 (red cartoon), that is buried in the nucleosome. The non-interacting parts are represented in grey cartoon. The figure was rendered by PyMol [47] using PDB structure 6tda [94].

**Figure 5 ijms-21-04134-f005:**
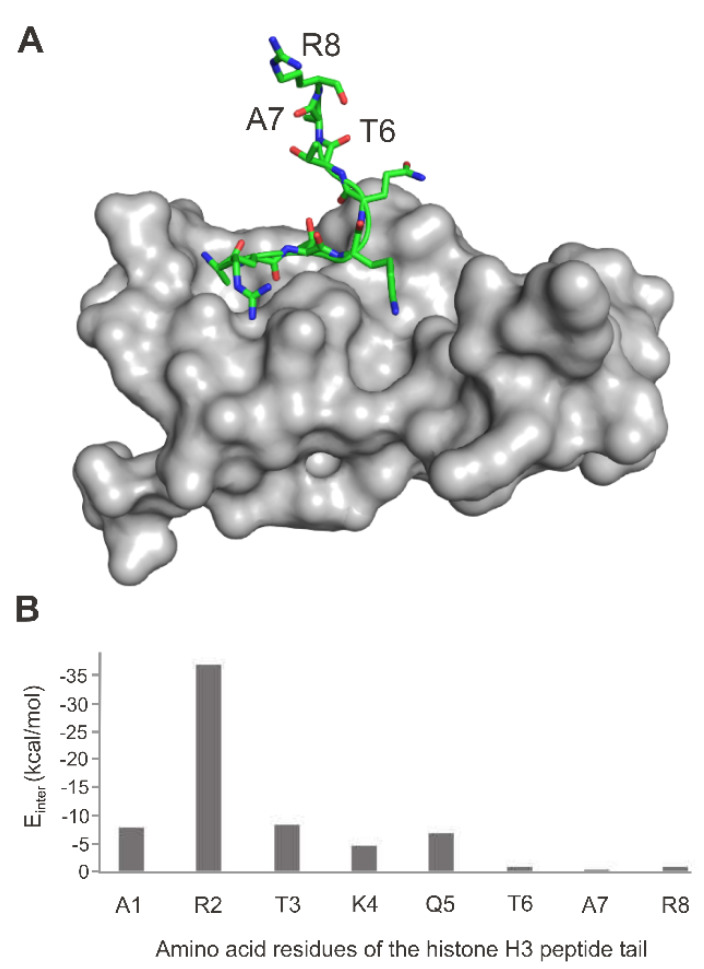
The structure (**A**) of the UHRF protein PHD finger (grey surface) in complex with histone H3 peptide tail (sticks, PDB ID 3sou). The figure was rendered by PyMol [47]. (**B**) Per residue peptide–protein interaction energies (Einter, bottom) were calculated after energy-minimization of the crystallographic complex. Einter values were calculated as a sum of Lennard-Jones and Coulomb interaction energies as described previously [99]. After the first 5 amino acid residues of the histone peptide, Einter diminishes as the last three residues interact with the bulk.

**Figure 6 ijms-21-04134-f006:**
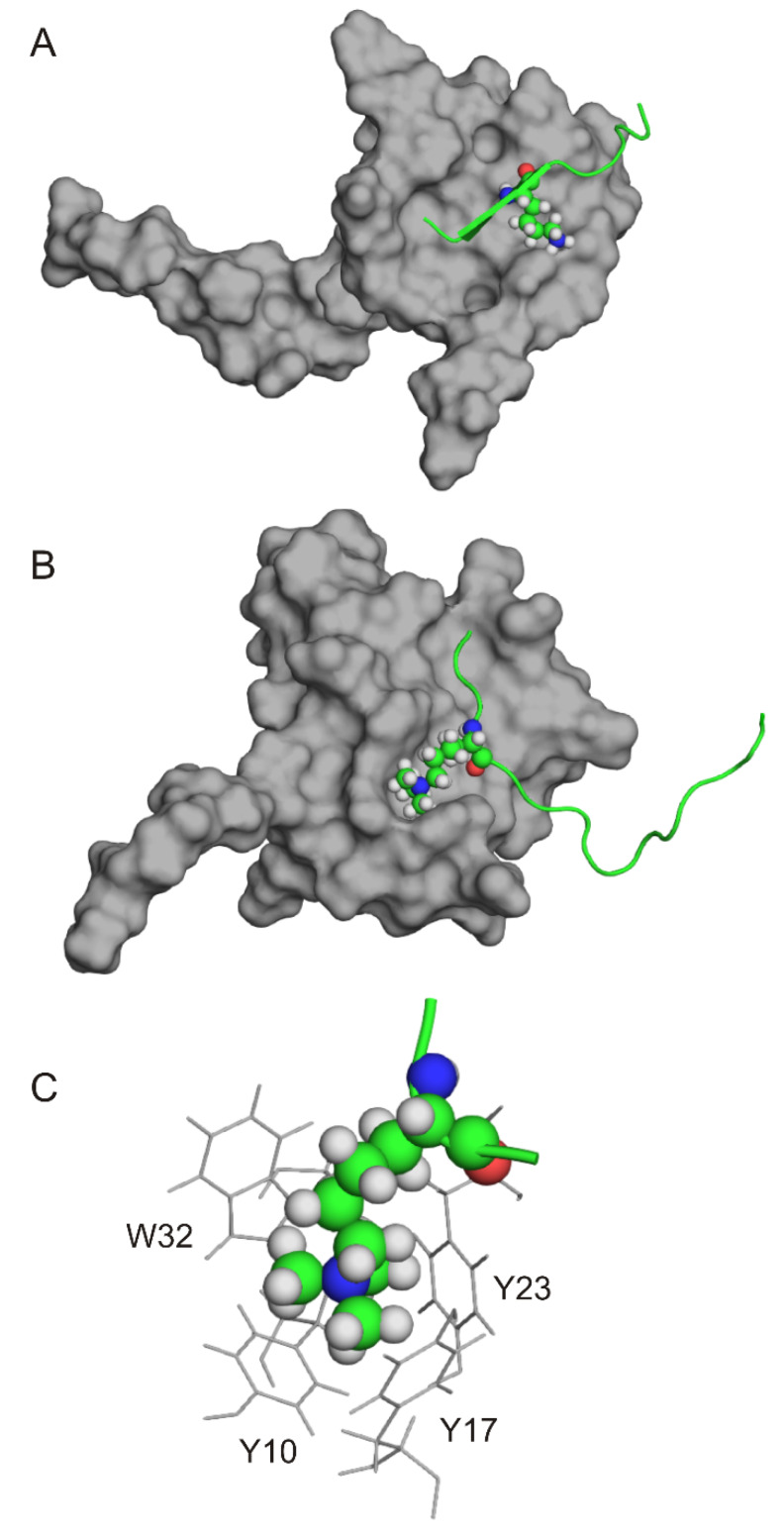
Binding of non-methylated (**A**) and tri-methylated (**B**,**C**) histone peptide tails (cartoon coils) to PHD fingers (grey surface). (**A**) The non-modified histone peptide tail with the H3K4me0 residue (spheres) binds the shallow surface of AIRE PHD finger (PDB ID 2ke1). (**B**) The histone peptide tail tri-methylated at H3K4me3 (spheres) binds to the aromatic cage of BPTF PHD finger (PDB ID 2fuu). (**C**) Close-up of the aromatic cage (BPTF PHD finger residues are in grey sticks) in complex with the tri-methylated lysine residue (H3K4me3, spheres). The figure was rendered by PyMol [47].

**Figure 7 ijms-21-04134-f007:**
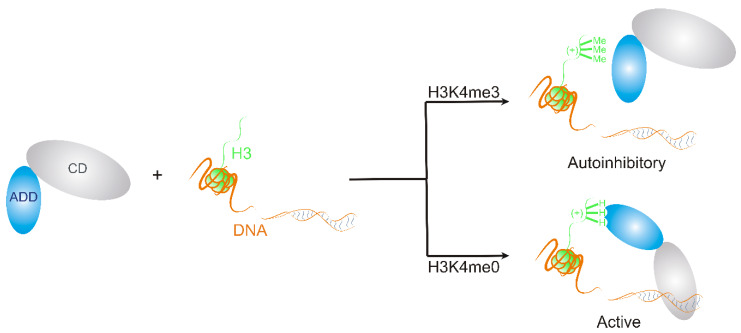
A schematic representation of the effect of H3K4me3 trimethylation on DNMT3A enzyme activity. In the presence of H3K4me3 PTM, the DNMT3A enzyme is in an autoinhibitory form. In the case of the non-methylated H3K4me0, the enzyme interacts with the DNA in an active form where the arrangements of its ADD and CD domains is different from that in the autoinhibitory form.

**Figure 8 ijms-21-04134-f008:**
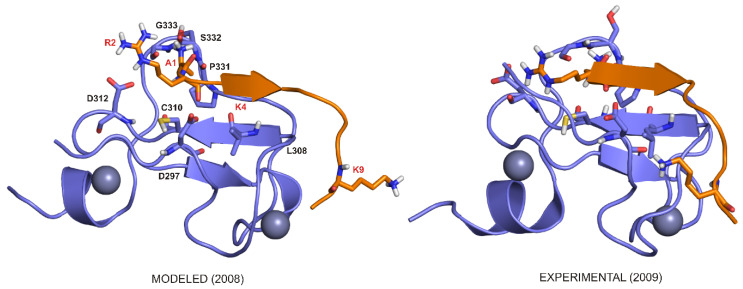
Comparison of modeled [130] and experimental [95] structures of histone H3K4me0 (orange cartoon) in complex with AIRE PHD finger (navy blue cartoon). Zinc ions and key residues are shown as navy blue spheres and labelled sticks, respectively. There is a good agreement between the modeled and the experimental structures both capturing the antiparallel ß-strand as binding conformation of histone H3K4me0.

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
