# Peer review of "Molecular Structure, Binding Affinity, and Biological Activity in the Epigenome"

_ijms, 2020, doi:10.3390/ijms21114134_

Round 1

Reviewer 1 Report

This is a well-written review article that will be useful to researchers in this important topic. I have some points I would like the authors to address.

  1. The term "SARs" is most often used in reference to small molecule drug design studies. Moreover, the title and abstract suggest there will be a strong focus on pharmacological studies. However, the manuscript overall is more broad and actually does not dive deeply into drug studies. The manuscript has more of a focus on proteins involved in epigenetics. This is fine, but I think the title and abstract should be reworded to be more consistent with the main focus of the manuscript. Perhaps the authors want to write that their paper focuses on epigenetic drug targets.
  2. For example, there is no mention of clinically used epigenetics drugs such as vorinostat or tazemetostat. There is also no mention of their target, EZH2, one of the most important proven epigenetic drug targets.
  3. There is too much discussion of very general research techniques such as cell viability assays or molecular mechanics calculations. I feel this is out of place for a review paper on epigenetics. Overall, the manuscript would benefit from tightening of the text and some editing and trimming.

Author Response

We thank the Reviewer for the positive comments and suggestions.

Comment 1

This is a well-written review article that will be useful to researchers in this important topic. I have some points I would like the authors to address.

The term "SARs" is most often used in reference to small molecule drug design studies. Moreover, the title and abstract suggest there will be a strong focus on pharmacological studies. However, the manuscript overall is more broad and actually does not dive deeply into drug studies. The manuscript has more of a focus on proteins involved in epigenetics. This is fine, but I think the title and abstract should be reworded to be more consistent with the main focus of the manuscript. Perhaps the authors want to write that their paper focuses on epigenetic drug targets.

Response 1

We used the term SAR in a more general context, not restricted to small molecule drugs, but also covering explanations of biological activities on the basis of structures of large molecules and their complexes. The review was aimed to show how the end-points (structure and activity) of SARs are uncovered, and connected in selected studies. According to the Reviewer’s suggestion we changed the wording from “pharmacological activity…” to “…biological activity…” not to emphasize the pharmacological point in the title, abstract, and section heads. A text on “proteins as epigenetic drug targets” is also mentioned to shift the focus of the new version of the abstract.

Comment 2

For example, there is no mention of clinically used epigenetics drugs such as vorinostat or tazemetostat. There is also no mention of their target, EZH2, one of the most important proven epigenetic drug targets.

Response 2

We agree that a list of current drugs was missing from the manuscript. A separate paragraph was inserted in the Conclusions with a brief overview of recent FDA approvals. EZH2 was also mentioned at this point.

Comment 3

There is too much discussion of very general research techniques such as cell viability assays or molecular mechanics calculations. I feel this is out of place for a review paper on epigenetics. Overall, the manuscript would benefit from tightening of the text and some editing and trimming.

Response 3

In agreement with the suggestion of the Reviewer, the text was trimmed at the cell viability assays (Section 2.2.2) and molecular mechanics (Section 3).

New text and the places of trimming were marked with yellow in the revised manuscript.

Reviewer 2 Report

This is a very nice and comprehensive review on structure of epigenome components and therapeutic targeting. It is quite comprehensive and I found it to nicely organized and easy to read. The manuscript should be closely read and checked for typos and places where writing could be improved. For example, the sentence on page 2, line 67 would read better as : DNA acetylation is controlled by two enzymatic families: 1) the histone lysine acetyltranferase (KAT); and 2) the histone deactylases (HDACs).

accept with minor revision

Author Response

We thank the Reviewer for the positive comments and suggestions.

Comment

This is a very nice and comprehensive review on structure of epigenome components and therapeutic targeting. It is quite comprehensive and I found it to nicely organized and easy to read. The manuscript should be closely read and checked for typos and places where writing could be improved. For example, the sentence on page 2, line 67 would read better as : DNA acetylation is controlled by two enzymatic families: 1) the histone lysine acetyltranferase (KAT); and 2) the histone deactylases (HDACs).

Response

We checked the manuscript and improved the text. The above-mentioned sentence was also corrected according to the Reviewer’s suggestions.

New text and the places of trimming were marked with yellow in the revised manuscript.